# Estimation of rate coefficients and branching ratios for gas-phase reactions of OH with aromatic organic compounds for use in automated mechanism construction

Michael E. Jenkin<sup>1,2</sup>, Richard Valorso<sup>3</sup>, Bernard Aumont<sup>3</sup>, Andrew R. Rickard<sup>4,5</sup>, Timothy J. 5 Wallington<sup>6</sup>

<sup>1</sup> Atmospheric Chemistry Services, Okehampton, Devon, EX20 4QB, UK

<sup>2</sup> School of Chemistry, University of Bristol, Cantock's Close, Bristol, BS8 1TS, UK

<sup>3</sup> LISA, UMR CNRS 7583, Université Paris Est Créteil et Université Paris Diderot, Institut Pierre Simon Laplace, 94010 Créteil, France

- <sup>4</sup> Wolfson Atmospheric Chemistry Laboratories, Department of Chemistry, University of York, YO10 5DD, UK
  <sup>5</sup> National Centre for Atmospheric Science, University of York, YO10 5DD, UK
  - <sup>6</sup> Research and Advanced Engineering, Ford Motor Company, SRL-3083, PO Box 2053, Dearborn, MI 48121-2053, USA

Correspondence to: Michael E. Jenkin (atmos.chem@btinternet.com)

- Abstract. Reaction with the hydroxyl (OH) radical is the dominant removal process for volatile organic compounds (VOCs) in the atmosphere. Rate coefficients for the reactions of OH with VOCs are therefore essential parameters for chemical mechanisms used in chemistry-transport models, and are required more generally for impact assessments involving estimation of atmospheric lifetimes or oxidation rates for VOCs. A structure-activity relationship (SAR) method is presented for the reactions of OH with aromatic organic compounds, with the reactions of aliphatic organic compounds considered in
- the preceding companion paper. The SAR is optimized using a preferred set of data including reactions of OH with 67 monocyclic aromatic hydrocarbons and oxygenated organic compounds. In each case, the rate coefficient is defined in terms of a summation of partial rate coefficients for H abstraction or OH addition at each relevant site in the given organic compound, so that the attack distribution is defined. The SAR can therefore guide the representation of the OH reactions in the next generation of explicit detailed chemical mechanisms. Rules governing the representation of the reactions of the
- product radicals under tropospheric conditions are also summarized, specifically the rapid reaction sequences initiated by their reactions with  $O_2$ .

## **1** Introduction

Aromatic hydrocarbons make a significant contribution to anthropogenic emissions of volatile organic compounds (VOCs), representing an important component of vehicle exhaust and other combustion emissions, and evaporative emissions of

30 petroleum and from industrial processes and solvent usage (e.g. Calvert et al., 2002; Passant, 2002). They are also emitted from sources that are either partially or wholly natural. They represent a significant proportion of VOC emissions from

biomass burning sources (e.g. Hays et al., 2002; Lewis et al., 2013), and are emitted substantially from vegetation (e.g. Misztal et al., 2015). An important contributor to these natural emissions is *p*-cymene (e.g. Helmig et al., 1998; Owen et al., 2001; Maleknia et al., 2007; Ulman et al., 2007), which is also formed as a degradation product of the reactive monoterpenes  $\alpha$ -terpinene,  $\alpha$ -phellandrene and  $\gamma$ -phellandrene (e.g. Berndt et al., 1996; Peeters et al., 1999; Aschmann et al., 2011). The

- 5 aromatic oxygenate, methyl chavicol (1-allyl-4-methoxybenzene), has also been reported to be emitted in large quantities from vegetation (Bouvier-Brown et al., 2009; Misztal et al., 2010), with evidence for a number of other aromatic oxygenates also reported (Misztal et al., 2015). It is well established that the gas phase degradation of VOCs in general plays a central role in the generation of a variety of secondary pollutants, including ozone and secondary organic aerosol, SOA (e.g. Haagen-Smit and Fox, 1954; Went, 1960; Andreae and Crutzen, 1997; Jenkin and Clemitshaw, 2000; Hallquist et al., 2009).
- By virtue of their generally high reactivity and emissions, the oxidation of aromatic compounds is believed to make an important contribution to the formation of ozone on local and regional scales (Derwent et al., 1996; Calvert et al., 2002), and to the formation of SOA in urban areas (e.g. Odum et al., 1997; Genter et al., 2017). The complete gas-phase oxidation of aromatic hydrocarbons proceeds via highly detailed mechanisms, producing a variety
- of intermediate oxidized organic products, some of which retain the aromatic ring (e.g. Calvert et al., 2002; Jenkin et al., 2003; Bloss et al., 2005). Reaction with the hydroxyl (OH) radical is generally the dominant or exclusive removal process for aromatic hydrocarbons, and makes a major contribution to the removal of aromatic oxygenates. Quantified rate coefficients for these reactions are therefore essential parameters for chemical mechanisms used in chemistry-transport models, and are required more generally for environmental assessments of their impacts, e.g. to estimate the kinetic component of ozone formation potentials (Jenkin et al., 2017). In addition to the total rate coefficient, quantification of the
- branching ratio for attack of OH at each site within a given compound is required for explicit representation of the subsequent oxidation pathways in chemical mechanisms. In the present paper, a structure-activity relationship (SAR) method is presented for the reactions of OH with aromatic organic compounds, with the reactions of aliphatic organic compounds considered in the preceding companion paper (Jenkin et al., 2018a). In each case, the rate coefficient is defined in terms of a summation of partial rate coefficients for H atom
- abstraction or OH addition at each relevant site in the given organic compound, so that the attack distribution is also defined. This is therefore the first generalizable SAR for reactions of OH with aromatic compounds that aims to capture observed trends in rate coefficients and the site-specificity of attack. Application of the methods is illustrated with examples in the Supplement.

The information is currently being used to guide the representation of the OH-initiation reactions in the next generation of

30 explicit detailed chemical mechanisms, based on the Generator for Explicit Chemistry and Kinetics of Organics in the Atmosphere, GECKO-A (Aumont et al., 2005), and the Master Chemical Mechanism, MCM (Saunders et al., 2003). It is therefore contributes to a revised and updated set of rules that can be used in automated mechanism construction, and provides formal documentation of the methods. To facilitate this, rules governing the representation of the reactions of the product radicals under tropospheric conditions are also summarized, specifically the rapid reaction sequences initiated by their reactions with  $O_2$ . The subsequent chemistry (e.g. reactions of peroxy radicals) will be considered elsewhere (Jenkin et al., 2018b).

# 2 Preferred kinetic data

- A set of preferred kinetic data has been assembled from which to develop and validate the estimation methods for the OH rate coefficients, as described in the companion paper (Jenkin et al., 2018a). The subset relevant to the present paper comprises 298 K data for 25 monocyclic aromatic hydrocarbons (with temperature dependences also defined in 13 cases); and 42 aromatic oxygenated organic compounds (with temperature dependences also defined in 7 cases). In one case (1,2diacetylbenzene), the preferred rate coefficient is an upper limit value. The information is provided as a part of the Supplement (spreadsheets SI\_6 and SI\_7). As described in more detail in Sect. 3.2, the oxygenates include compounds containing a variety of oxygenated substituent groups that are prevalent in both emitted VOCs and their degradation products, namely -OH, -C(OH)<, -C(=O)-, -O- , -C(=O)O- and -NO<sub>2</sub> groups. For a core set of 11 reactions, the preferred kinetic data are based on the evaluations of the IUPAC Task Group on Atmospheric Chemical Kinetic Data Evaluation (http://iupac.pole-ether.fr/). The remaining values are informed by recommendations from other key evaluations with complementary coverage (e.g. Atkinson and Arey, 2003; Calvert et al., 2011), and have been revised and expanded
- following review and evaluation of additional data not included in those studies (as identified in spreadsheets SI\_6 and SI\_7).

#### 3 Kinetics and branching ratios of initiation reactions

The reaction of OH with a given aromatic compound can occur by both addition of OH to the aromatic ring and by abstraction of an H-atom from a C-H or O-H bond in a substituent group. The estimated rate coefficient is therefore given by  $k_{calc} = k_{add} + k_{abs}$ ,

- where  $k_{add}$  and  $k_{abs}$  are summations of the partial rate coefficients for OH addition and H-atom abstraction for each attack position in the given aromatic compound. Based on reported data for the reaction of OH with benzene, abstraction of H-atoms from the aromatic ring itself is assumed to be negligible under atmospheric conditions (e.g. see Calvert et al., 2002). A method for estimating rate coefficients for OH addition to the aromatic ring ( $k_{add}$ ), and the distribution of attack, is described in the sections that follow. The estimation of rate coefficients for H-atom abstraction from substituent groups ( $k_{abs}$ ) follows the
- methods described in the companion paper (Jenkin et al., 2018a), which are mainly based on updating and extending the widely applied method of Kwok and Atkinson (1995). For C-H bonds, the estimated rate coefficients are thus generally based on a summation of rate coefficients for H-atom abstraction from the primary (-CH<sub>3</sub>), secondary (-CH<sub>2</sub>-) and tertiary (-CH<) groups which are calculated as follows:</p>

$$k(CH_3-X) = k_{prim}F(X)$$
<sup>(1)</sup>

$$30 \quad k(X-CH_2-Y) = k_{sec} F(X) F(Y) \tag{2}$$

 $k_{\text{prim}}$ ,  $k_{\text{sec}}$  and  $k_{\text{tert}}$  are the respective group rate coefficients for abstraction from primary, secondary and tertiary groups for a reference substituent; and F(X), F(Y) and F(Z) are factors that account for the effects of the substituents X, Y and Z. The reference substituent is defined as "-CH<sub>3</sub>", such that F(-CH<sub>3</sub>) = 1.00 (Atkinson, 1987; Kwok and Atkinson, 1995). As

- described in detail in the companion paper (Jenkin et al., 2018a), a number of fixed rate coefficients are also defined for Hatom abstraction from O-H bonds in hydroxy, hydroperoxy and carboxyl groups; and for C-H bonds in a series of formyl groups, and adjacent to -O- linkages in ethers. The values of these rate coefficients are assumed to be independent of the identity of neighbouring substituent groups. The methods summarized above are extended in the present work to include rate coefficients and neighbouring group substituent factors for H-atom abstraction from carbon and oxygen atoms adjacent to
- aromatic rings.

For aromatic compounds containing an unsaturated substituent, the addition of OH to C=C bonds in the substituent group can also occur. The treatment of these reactions is described in Sect. 3.1.3.

## 3.1 Aromatic hydrocarbons

## 3.1.1 Methyl-substituted aromatic hydrocarbons

- The set of preferred kinetic data contains rate coefficients for the reactions of OH with 12 methyl-substituted aromatic hydrocarbons possessing between one and six methyl substituents. This class is the most comprehensively studied, with room temperature data covering all possible methyl-substituted isomers. Although rate coefficients for this class of compound do not therefore need to be estimated, the SAR described below aims to rationalize the variation of reactivity from one compound to another, and to provide a method of estimating the OH attack distributions that can be applied in automated mechanism
- generation.
  - The contribution of H-atom abstraction to the total rate coefficient is known to be minor at temperatures relevant to the atmosphere for methyl-substituted aromatics (e.g. Calvert et al., 2002; Loison et al., 2012; Aschmann et al., 2013). The temperature-dependent reference substituent factor for a phenyl group, F(-Ph1) (see Table 1), was set so that the H-atom abstraction rate coefficient for the methyl group in toluene matches the IUPAC recommendation, i.e.  $2.5 \times 10^{-11} \exp(-1270/T)$
- cm<sup>3</sup> molecule<sup>-1</sup> s<sup>-1</sup> (IUPAC, 2017a). The branching ratios reported in the above studies suggest that H-atom abstraction is slightly more efficient for methyl groups in some polymethyl-substituted aromatics, particularly for hexamethylbenzene (Loison et al., 2012), probably reflecting an additional stabilizing effect on the resonant product radical. The data were thus found to be reasonably well described by assigning a further activation factor of exp(140/T) (equating to a value of 1.6 at 298 K) for each additional methyl group positioned *ortho* or *para* to the abstraction group. The resultant estimated branching ratios for H-atom
- abstraction are discussed further below.

The current estimation method defines site-specific parameters for addition of OH to each carbon atom in the aromatic ring. As shown in Table 2,  $k_{arom}$  is used to represent addition of OH to an unsubstituted carbon, and  $k_{ipso}$  is used to represent addition of

OH to a methyl-substituted carbon. The total rate coefficient for OH addition is then given by a summation of the partial rate coefficients for each of the six attack positions,

$$k_{\rm add} = \Sigma k \, \mathrm{F}(\Phi) \tag{4}$$

where k is either  $k_{arom}$  or  $k_{ipso}$  and  $F(\Phi)$  is a factor that accounts for the effect of the combination of methyl substituents in the molecule in terms of their positions (i.e. *ortho-, meta-* or *para-*) relative to each OH addition location.

As shown in Table 3, the dataset was described in terms of 11 substituent factors, representing the effects of between one and five methyl substituents. Based on the results of previous assessments (e.g. see Calvert et al., 2002), the number of parameters was limited by assuming that *ortho-* and *para-* substituents have the same influence, whether individually or in combinations. Examples of rate coefficient calculations using these parameters are given in the Supplement.

- The values of the  $F(\Phi)$  factors in Table 3 and  $k_{ipso}$  were varied iteratively to minimize the summed square deviation,  $\Sigma((k_{calc}-k_{obs})/k_{obs})^2$  at 298 K for the set of methyl-substituted aromatic hydrocarbons. Within the context of previous appraisals (e.g. Calvert et al., 2002 and references therein), the resultant values show some consistent trends, with *ortho-* and *para-* substituents being significantly more activating than *meta-* substituents. It is also interesting to note that the elevation in  $k_{ipso}$  relative to  $k_{arom}$  (i.e. a factor of 1.4) is identical to the activating influence of a lone *meta-* substituent, which is also consistent with previous
- assumptions (e.g. Calvert et al., 2002). Increasing the number of substituents has a generally increasing activating impact, although the highest value was returned for F(o-,o-,p-), i.e. for three substituents in the most activating positions, with this value being determined by the observed rate coefficients for 1,3,5-trimethylbenzene and 1,2,3,5-tetramethylbenzene. A correlation of the optimized values of  $k_{calc}$  with  $k_{obs}$  at 298 K is shown in Fig. 1. The estimation method reproduces all the observed values to within 5 %.
- The estimated contributions of H-atom abstraction from the methyl substituents in the series of aromatics hydrocarbons are compared with those reported, in Table 4. The values confirm that rate coefficients assigned to these reactions in Table 1 provide a reasonable description for the complete dataset of methyl-substituted aromatics.

There have been no direct experimental determinations of the branching ratios for OH addition to methyl-substituted aromatic rings, although a number of Density Functional Theory (DFT) studies have been reported for toluene, *m*-xylene, *p*-xylene and

- 1,2,4-trimethylbenzene (Suh et al., 2002; Fan et al., 2006; 2008; Huang et al., 2011; Wu et al., 2014; Li et al., 2014). As shown in Table 5, the attack distributions of OH predicted by the partial rate coefficients determined from the present method are generally consistent with those reported in the theoretical studies, providing a level of independent support for the method developed here. The distributions for toluene and *p*-xylene are in good agreement with those reported in the DFT studies, with those for 1,2,4-trimethyl benzene also being in reasonable agreement. For *m*-xylene, the major channels (i.e. addition at positions (2) and (3)) are
- consistent with those reported by Fan et al. (2008) and Huang et al. (2011), although their relative importance is reversed. The present method predicts addition at position (3) to be more important because of its greater degeneracy, whereas the DFT studies predict that this is outweighed by a much stronger activating influence of the two *ortho* substitutions on position (2) compared with that of the *ortho* and *para* substitutions on position (3). Conversely, the opposite appears to be the case for 1,2,4-trimethyl

benzene, where the DFT study of Li et al. (2014) calculates position (5) (with *ortho-* and *para-* substitutions) to be favoured over position (4) (with two *ortho-* substitutions), despite both sites being singly degenerate in that case.

Temperature-dependent recommendations are available for benzene and 10 methyl-substituted aromatics in Arrhenius format (k = A.exp(-(E/R)/T)) (see spreadsheet SI\_6). These were used to provide optimized temperature coefficients ( $B_{F(\Phi)}$ ) and preexponential factors ( $A_{F(\Phi)}$ ) for the set of OH addition substituent factors given in Table 3. Optimization was achieved by calculating values of k at even 1/T intervals over the recommended temperature range for each aromatic, and determining a composite E/R value from a least squares linear regression of the data on an Arrhenius (i.e. ln(k) vs. 1/T) plot. The 11 values of values of  $B_{F(\Phi)}$  in Table 3 were varied to minimize the summed square deviation in the composite temperature coefficients,  $\Sigma((E/R)_{calc}-(E/R)_{obs})^2$ . The resultant  $(E/R)_{calc}$  values are compared with the recommended  $(E/R)_{obs}$  values in the lower panel of Fig.

10 1 (see also Fig. S1). The values of  $A_{F(\Phi)}$  were automatically returned from the corresponding optimized  $B_{F(\Phi)}$  and  $F(\Phi)_{298 \text{ K}}$  values.

#### **3.1.2 Higher alkyl-substituted aromatic hydrocarbons**

The set of preferred kinetic data contains rate coefficients for a further eight alkyl-substituted aromatic hydrocarbons, namely ethylbenzene, *n*-propylbenzene *i*-propylbenzene, *t*-butylbenzene, *o*-ethyltoluene, *m*-ethyltoluene, *p*-ethyltoluene and *p*-cymene. Information on H-atom abstraction from this series of compounds is limited to the study of *p*-cymene (4-*i*-propyltoluene) reported

- by Aschmann et al. (2010) and Bedjanian et al. (2015), who determined a total branching ratio for H-atom abstraction of about 20 %, with about 15 % from the -CH< group in the *i*-propyl substituent (see Table 4). Use of the aromatic substituent factors appropriate to H-atom abstraction from  $\alpha$  -CH<sub>3</sub> groups (i.e. F(-Ph1) in Table 1) would clearly lead to a gross overestimation for *p*-cymene (i.e. about 34 % from the -CH< group in the *i*-propyl substituent and a total of about 39 %), and also unreasonably large contributions in the other compounds identified above. Based on the *p*-cymene data, a substituent factor of 1.0 is assigned to F(-
- Ph2), representing H-atom abstraction from a substituent  $\alpha$  -CH< group, and also applied to abstraction from an  $\alpha$  -CH<sub>2</sub>- group in  $\geq$  C<sub>2</sub> substituents (see Table 1). As for the -CH<sub>3</sub> groups discussed above, the further activation factor of exp(140/T) (equating to a value of 1.6 at 298 K) is applied for each additional alkyl group positioned *ortho-* or *para-* to the abstraction group. For *p*cymene, this results an estimated total branching ratio for H-atom abstraction of 22.4 %, with 16.2 % from the -CH< group in the *i*-propyl substituent (see Table 4), in good agreement with the observations of Aschmann et al. (2010) and Bedjanian et al. (2015).
- It is noted that the value of 1.0 assigned to F(-Ph2) at 298 K is unchanged from that previously reported by Kwok and Atkinson (1995) for phenyl groups in general.

The methyl group substituent factors in Table 3 provide a reasonable first approximation for the effects of the higher alkyl groups on OH addition rate coefficients, and use of those factors leads to a set of estimated rate coefficients that are all within 30 % of the observed values for the current set of eight higher alkyl-substituted aromatic hydrocarbons. On the whole, however, this results in

a slight overestimation of the rate coefficients. Table 6 shows a set of adjustment factors for non-methyl substituents,  $R(\Phi)$ , that represent corrections to the values of  $F(\Phi)$  in Table 3 (and to  $k_{ipso}$ , when appropriate), such that:

$$k_{\text{add}} = \Sigma k F(\Phi) R(\Phi)$$

(5)

These result in a generally improved agreement, with deviations from the observed rate coefficients of  $\leq 16$  % (see Fig. 1). For the present set of compounds, these adjustment factors are only defined for the impacts of *ortho-* and *para-* substitutions, as adjustments for *meta-* and *ipso-* groups appeared to result in more subtle benefits. In principle, a value of R( $\Phi$ ) should be applied for each higher alkyl group in the molecule, although none of the current set contains more than one higher alkyl substituent. The

- 5 factors appear to show a deactivating effect (relative to that of methyl) that increases with the size of the alkyl group, with this being qualitatively consistent with information reported in previous appraisals (e.g. see Calvert et al., 2002). It is emphasized, however, that the adjustment factors are derived from the analysis of a very small dataset, with some factors based on reported data for a single compound. Clearly, further systematic kinetic studies of higher alkyl-substituted aromatics would be of benefit. Similarly to above, there have been no direct experimental determinations of branching ratios for OH addition to higher alkyl-
- 10 substituted aromatics, although Huang et al. (2010) have reported a DFT study for ethylbenzene, and Alarcón et al. (2014) for *p*cymene. As shown in Table 5, the attack distributions of OH predicted by the partial rate coefficients determined from the present method agree reasonably well with those reported.

Temperature-dependent studies are only available for *p*-cymene (Alarcón et al., 2014; Bedjanian et al., 2015), resulting in a recommended value of E/R = -640 K. The parameters discussed above are unable to recreate this temperature dependence, and

- logically return a temperature dependence comparable to that of the structurally similar compound *p*-xylene, for which the recommended *E/R* = -160 K. It was found that this discrepancy could be resolved by applying a temperature dependent value of R<sub>*i*-pr</sub>(*o*-) = R<sub>*i*-pr</sub>(*p*-) = 0.029.exp(1000/T) (see Table 6 comment (d)). This results in *i*-propyl groups becoming more activating relative to methyl groups as the temperature is lowered, with values of R<sub>*i*-pr</sub>(*o*-) and R<sub>*i*-pr</sub>(*p*-) >1 at temperatures below about 280 K. The DFT calculations of Alarcón et al. (2014) provide some support for this trend for R<sub>*i*-pr</sub>(*o*-). Provisional temperature 20 dependences are also suggested for the other R<sub>alkyd</sub>(*o*-) and R<sub>alkyd</sub>(*p*-) values (see Table 6 comments), although it is again
- dependences are also suggested for the other  $R_{alkyl}(o)$  and  $R_{alkyl}(p)$  values (see Table 6 comments), although it is again emphasized that these parameters are generally based on very limited information.

# 3.1.3 Alkenyl-substituted aromatic hydrocarbons

The set of preferred kinetic data contains rate coefficients for the reactions of OH with four alk-1-enyl (or vinyl) substituted aromatic hydrocarbons, namely styrene (ethenylbenzene),  $\alpha$ -methylstyrene (*i*-propenylbenzene),  $\beta$ -methylstyrene (propenylbenzene) and  $\beta$ , $\beta$ -dimethylstyrene (2-methylpropenylbenzene). Experimental and theoretical information for the most studied compound, styrene, is consistent with the reaction occurring predominantly by addition of OH to the ethenyl substituent (Bignozzi et al., 1981; Tuazon et al., 1993; Cho et al., 2014). However, unlike the trends in rate coefficients for aliphatic alkenes (see Sect. 4.1.1 of Jenkin et al., 2018a), the presence of the alkyl substituents on the alkene group in the series of styrenes does not apparently enhance the reactivity, with very similar 298 K rate coefficients reported for styrene,  $\alpha$ -methylstyrene and  $\beta$ methylstyrene, and a reduction in reactivity for the most substituted compound,  $\beta$ , $\beta$ -dimethylstyrene. A fixed rate coefficient,  $k_{C=C-Ph} = 9.8 \times 10^{-12} \exp(530/T) \text{ cm}^3 \text{ molecule}^{-1} \text{ s}^{-1}$ , is therefore provisionally assigned to addition of OH to alk-1-enyl (vinyl) substituents, based on the preferred value for styrene at 298 K, and the value of *E/R* calculated by Cho et al. (2014). The reaction is assumed to occur exclusively by addition to the  $\beta$  carbon in the substituent group, because this forms a resonance-stabilized radical. Accordingly, the presence of an alk-1-envl (vinyl) substituent is assumed to result in complete deactivation of OH addition to the aromatic ring (see Table 6).

The addition of OH to more remote C=C bonds in substituent groups in alkenyl-substituted aromatic hydrocarbons is expected to 5 be well described by the methods described in the companion paper (Jenkin et al., 2018a), which update and extend the methods reported by Peeters et al. (2007) for alkenes and dienes. However, there are currently no data to test this assumption. In these cases, it is suggested that a default value of  $R(\Phi) = 1.0$  for the remote alkenyl group is applied for addition of OH to the aromatic ring.

# 3.2 Monocyclic aromatic oxygenates

- 10 The preferred 298 K data include rate coefficients for reactions of OH with 42 aromatics containing a variety of oxygenated substituent groups, which were used to extend the methods described above for estimating rate coefficients for aromatic hydrocarbons. Rate coefficients for H atom abstraction from the oxygenated groups are generally represented using the methods applied to aliphatic oxygenates (Jenkin et al., 2018a), in conjunction with the values of F(X) given in Table 1, where appropriate; but with specific parameters defined for abstraction from -OH and -C(=O)H substituents (see Sects. 3.2.1 - 3.2.3). For addition of 15
- OH to the aromatic ring, the influences of the oxygenated substituents are described by the set of adjustment factors,  $R(\Phi)$ , given in Table 6. As for the higher alkyl substituents discussed in Sect. 3.1.2, these represent corrections to the values of  $F(\Phi)$  in Table 3, and to k<sub>ipso</sub> in Table 2, and are applied for each oxygenated substituent in the given molecule. They thus describe the effect of the oxygenated substituent relative to that of a -CH<sub>3</sub> group in the same position. In many cases, values of  $R(\Phi)$  are derived from the analysis of a limited number of compounds containing the relevant substituent, with some based on reported data for a single
- 20 compound, as summarized in the notes to Table 6. However, the values for -OH, -C(=O)H and -NO<sub>2</sub> are based on analysis of larger sets of compounds, as described in following subsections. With the exception of three catechols, the values of  $R(\Phi)$  in Table 6 are determined from sets of compounds containing only one of the relevant oxygenated substituent. As a result, extrapolation of the method to compounds containing several activating substituents can result in unreasonably high estimated rate coefficients (i.e. exceeding the bimolecular collision rate). An upper limit rate coefficient,  $k_{calc} = 3.0 \times 10^{-10} \text{ cm}^3 \text{ molecule}^{-1} \text{ s}^{-1}$ <sup>1</sup>, is therefore imposed. Further data for aromatics containing multiple oxygenated substituents are clearly required to allow the 25

method to be tested and refined.

# 3.2.1 Phenols and catechols

The contribution of H-atom abstraction from the -OH substituent in phenolic compounds has generally been inferred from the measured yields of nitrophenolic products, under conditions when the intermediate phenoxy radicals are expected to react

30 predominantly with NO<sub>2</sub>. Based on the nitrophenol yields reported for phenol and the set of cresol isomers by Atkinson et al. (1992), Olariu et al. (2002), Berndt and Böge (2003) and Coeur-Tourneur et al. (2006), an average rate coefficient,  $k_{abs(Ph-OH)} = 2.6$  $\times 10^{-12}$  cm<sup>3</sup> molecule<sup>-1</sup> s<sup>-1</sup>, is assigned to this abstraction reaction at 298 K. This is about a factor of 20 greater than estimated for abstraction from -OH groups in aliphatic compounds (Jenkin et al., 2018a), which can be attributed to the resonance stabilization of the product phenoxy radicals. This suggests that the value of  $k_{abs(Ph-OH)}$  may therefore be influenced by the presence of other substituents on the aromatic ring. This cannot be confirmed unambiguously from the reported dataset for phenols and cresols, although the presence of the *ortho*- NO<sub>2</sub> group in 2-nitrophenols appears to have a significant deactivating effect (see Sect. 3.2.3).

- There is currently insufficient information to allow a full appraisal of the effects of the variety of possible substituents groups on H-atom abstraction from -OH (or other) substituents. In the present work, therefore, the above value of  $k_{abs(Ph-OH)}$  is applied, unless the compound contains either an *ortho*- NO<sub>2</sub> group or (by inference) a *para*- NO<sub>2</sub> group.  $k_{abs(Ph-OH)}$  is assumed to be independent of temperature over the atmospheric range, which is consistent with the provisional temperature dependence expressions suggested by Atkinson (1989), inferred from extrapolation of higher temperature data for phenol and *o*-cresol.
- The values of  $R_{OH}(\Phi)$  in Table 6 were varied iteratively to minimize the summed square deviation,  $\Sigma((k_{calc}-k_{obs})/k_{obs})^2$  at 298 K for phenol, 14 methyl-substituted phenols, catechol and two methyl-substituted catechols. The resultant values of  $k_{calc}$  agree reasonably well with  $k_{obs}$  for the complete set of compounds (see Fig. 2), with particularly good agreement for the the more substituted phenols and the catechols. Although the agreement is less good for the smaller, less reactive compounds (particularly for phenol,  $k_{calc}/k_{obs} \approx 0.6$ , and *p*-cresol,  $k_{calc}/k_{obs} \approx 0.7$ ), the values of  $R_{OH}(\Phi)$  are considered appropriate for wider application to
- multifunctional aromatic compounds containing -OH substituents for which there is currently no information. Temperature dependent data are currently limited to phenol and the cresol isomers. Use of the temperature dependent factors given in Table 6 allows a reasonable representation of observed preferred temperature dependences, as shown in the lower panel of Fig. 2 (see also, Fig. S2).

The attack distributions predicted by the optimized parameters recreate some of the features inferred from reported experimental

- studies for phenol and cresols (e.g. Olariu et al., 2002), initiating routes to the observed formation of catechols (1,2dihydroxyarenes), benzoquinones and nitrophenols (see Sect. 4.2). As shown in Table 6, comparable values of  $R_{OH}(\Phi)$  for each attack position are required to recreate the observed kinetics for the complete set of phenolic compounds. As a result, the -OH substituent retains the greater *ortho-* and *para-* directing influence discussed above for the reference substituent, -CH<sub>3</sub>. The optimized parameters therefore predict significant formation of catechols from the oxidation of mono-phenols (resulting from
- ortho- attack), qualitatively consistent with the results of the experimental studies. However, the optimized ortho- directing influence of the -OH substituent is still insufficient to recreate the observed dominant (65-80 %) formation of catechol products, reported for phenols and cresols (e.g. Olariu et al., 2002). Noting that the product studies mainly consider the smaller compounds for which the parameter optimization procedure works least well, this may be indicative of the contribution of ortho- attack of OH being underestimated for these compounds, but with the method being reasonable for wider application to more substituted
- aromatic products containing -OH substituents. It is generally recommended that attack distributions (and rate coefficients) based on the results of experimental studies are applied where evaluated information is available, as presented specifically for phenol and the cresol isomers in Sect. S3.

## 3.2.2 Benzaldehydes

The set of preferred kinetic data contains rate coefficients for benzaldehyde, three methyl-substituted benzaldehydes and six dimethyl-substituted benzaldehydes. In addition, preferred data are included for phthaldialdehyde (1,2-diformylbenzene) and 2-acetylbenzaldehyde, and an upper limit rate coefficient for the related compound 1,2-diacetylbenzene, based on Wang et al.

- 5 (2006). The data show that the presence of methyl substituents in the benzaldehydes increases the OH reactivity systematically. It is generally accepted that abstraction of the H-atom from the formyl (-C(=O)H) substituent is the dominant pathway for benzaldehyde, and this has been estimated to account for about 96 % of the reaction at 298 K in the DFT study of Iuga et al. (2008). As discussed previously (e.g. Thiault et al., 2002; Clifford et al., 2005; Clifford and Wenger, 2006), the activating effect of the methyl substituents may therefore result from an increasing contribution of OH addition and/or from an activating influence on the abstraction rate from the formyl substituent.
- Initially, it was assumed that the rate coefficient for H-atom abstraction from the formyl group,  $k_{abs(Ph-C(O)H)}$ , remains constant for the complete series of compounds. Values of  $k_{abs(Ph-C(O)H)}$ , and of a set of adjustment factors for OH addition,  $R_{C(O)H}(\Phi)$ , were varied iteratively to minimize  $\Sigma((k_{calc}-k_{obs})/k_{obs})^2$  at 298 K, leading to a set of parameter values given in Sect. S4 (Table S4). These predict that the contribution of H-atom abstraction from benzaldehyde is 86 %, decreasing to 36–46 % for the dimethylbenzaldehyde isomers. Although this is consistent with a major contribution for benzaldehyde, the predicted value is
- significantly lower than the 96 % calculated for H-atom abstraction by Iuga et al. (2008). With the reasonable assumption that the values of  $R_{C(O)R}(\Phi)$  for -C(=O)H substituents can also be applied more generally to -C(=O)R substituents, the estimated rate coefficient for 1,2-diacetylbenzene also exceeds the reported upper limit value by more than a factor of two. This suggests that these optimized parameters also significantly overestimate OH addition to the aromatic ring.
- An alternative procedure was therefore adopted in which the contribution of H-atom abstraction from the -C(=O)H group in benzaldehyde was constrained to 96 % at 298 K (providing a reference value of  $k_{abs(Ph-C(O)H)} = 1.21 \times 10^{-11}$  cm<sup>3</sup> molecule<sup>-1</sup> s<sup>-1</sup>); and the values of  $R_{C(O)H}(\Phi)$  were varied to reproduce the total rate coefficient for benzaldehyde, leading to the (strongly deactivating) values presented in Table 6. The activating influence of the methyl substituents is then partly accounted for by increases in the OH addition rate coefficients, but also requires H-atom abstraction from the -C(=O)H group to be enhanced. Based on
- optimization to the complete set of rate coefficients, the data were found to be well described by assigning activation factors of exp(115/T) (equating to a value of 1.47 at 298 K) for a methyl group positioned *ortho* to the -C(=O)H group group, and exp(78/T) (equating to a value of 1.30 at 298 K) for a methyl group positioned either *meta* or *para* to the -C(=O)H group (with these factors also assumed to apply to other alkyl groups). A correlation of the optimized values of  $k_{calc}$  with  $k_{obs}$  at 298 K is shown in Fig. 2, with the estimation method reproducing all the observed values to within 10 %. Based on this approach, H-
- 30 atom abstraction from the -C(=O)H group remains the most important route, decreasing from 96 % for benzaldehyde to 76–88 % for the dimethylbenzaldehyde isomers. The optimized parameters also provide a reasonable description of the data for phthaldialdehyde (1,2-diformylbenzene) and 2-acetylbenzaldehyde (identified as aromatic dicarbonyls in Fig. 2), and an

estimated rate coefficient for 1,2-diacetylbenzene ( $3.8 \times 10^{-13}$  cm<sup>3</sup> molecule<sup>-1</sup> s<sup>-1</sup>) that is consistent with the reported upper limit value ( $< 1.2 \times 10^{-12}$  cm<sup>3</sup> molecule<sup>-1</sup> s<sup>-1</sup>).

Temperature-dependent data are only available for benzaldehyde. Within the constraints of the approach described above, this was used to provide the optimized temperature dependence expression,  $k_{abs(Ph-C(O)H)} = 5.23 \times 10^{-12} \exp(250/T) \text{ cm}^3 \text{ molecule}^{-1} \text{ s}^{-1}$ .

#### 5 **3.2.3** Nitroarenes and nitrophenols

The set of preferred kinetic data contains rate coefficients for the reactions of OH with a number of nitro-substituted aromatics, namely nitrobenzene, 1-methyl-3-nitrobenzene, 2-nitrophenol and four methyl-substituted 2-nitrophenols. These data were used to optimize the values of  $R_{NO2}(\Phi)$  in Table 6, with the values of  $R_{OH}(\Phi)$  determined in Sect. 3.2.1 applied where appropriate. During this procedure, it became clear that the value of  $k_{abs(Ph-OH)}$  (also optimized in Sect. 3.2.1) substantially

- overestimates the importance of H-atom abstraction from the -OH substituent in 2-nitrophenols. The data therefore suggest that an *ortho*- NO<sub>2</sub> group (and possibly also a *para*- NO<sub>2</sub> group) has a strong deactivating effect on this reaction, and the data were best described by reducing its rate by at least an order of magnitude, compared with  $k_{abs(Ph-OH)}$ . It was therefore assumed that the rate coefficient previously assigned to -OH groups in aliphatic compounds,  $k_{abs(-OH)} = 1.28 \times 10^{-12} \exp(-660/T) \text{ cm}^3$  molecule<sup>-1</sup> s<sup>-1</sup>, applies when the aromatic ring is deactivated by the presence of an NO<sub>2</sub> group *ortho*- or *para*- to the -OH substituent. As
- indicated above, additional information is clearly required to allow a full appraisal of the effects of substituent groups on H-atom abstraction from -OH (or other) substituents in aromatic compounds. The optimized values of  $R_{NO2}(\Phi)$  in Table 6 indicate that NO<sub>2</sub> substituents also strongly deactivate addition of OH to the aromatic

ring. As shown in Fig. 2, the resultant values of  $k_{calc}$  agree well with  $k_{obs}$  for the complete set of nitro-substituted compounds identified above.

## 20 4 Reaction of O<sub>2</sub> with OH-aromatic adducts and subsequent chemistry

# 4.1 OH-aromatic hydrocarbon adducts

A method has been developed to describe the chemistry initiated by reaction of  $O_2$  with the OH-aromatic adducts formed from the addition of OH radicals to aromatic hydrocarbons. Theoretical studies have shown that these reactions, and the subsequent reaction sequences, can be highly complex, involving the participation of geometrical isomers of very different reactivities (e.g.

- Raoult et al., 2004; Glowacki et al., 2009; Wu et al., 2014; Li et al., 2014; Pan and Wang, 2014; Vereecken, 2018a). The present method does not include the level of detail established in these studies, but aims to provide an empirically-optimized reaction framework incorporating the main features of the mechanisms, as reported in both laboratory and theoretical work. The reactions of the OH-aromatic hydrocarbon adducts with  $O_2$  are represented to react either by direct α- H atom abstraction, forming HO<sub>2</sub> and a hydroxyarene (phenolic) product, or by β- O<sub>2</sub> addition to the aromatic ring at each of the two carbon atoms
- adjacent to the -OH substituent to produce  $\beta$ -hydroxy cyclohexadienylperoxy radicals (as illustrated in Fig. 3), such that the

overall rate coefficient is given by  $k_{abs-O2} + k_{add-O2(1)} + k_{add-O2(2)}$ . The H atom abstraction reaction is unavailable for adducts formed from OH addition *ipso*- to an alkyl substitution. There is some evidence for a "dealkylation" pathway from such adducts (e.g. Noda et al., 2009), but this is not currently represented owing to conflicting evidence on its significance (e.g. Aschmann et al., 2010; Loison et al., 2012). In practice, the  $\beta$ - O<sub>2</sub> addition pathways are reversible, such that each value of  $k_{add-O2}$  specifically

quantifies the effective irreversible component of the reaction that results in onward removal of the given cyclohexadienylperoxy radical (IUPAC, 2017b; 2017c).

The value of  $k_{abs-O2}$  and the reference value of  $k_{add-O2}^{\circ}$  for the benzene system (see Table 7) are informed by the calculations of Raoult et al. (2004), but adjusted to give a total rate coefficient of  $\sim 2.1 \times 10^{-16}$  cm<sup>3</sup> molecule<sup>-1</sup> s<sup>-1</sup> at 298 K for (the irreversible component of) the reaction of HOC<sub>6</sub>H<sub>6</sub> with O<sub>2</sub>, as recommended by IUPAC (2017b); and a yield of phenol of  $\sim 53$  %, which is also consistent with the literature. The value of  $k_{abs-O2}$  is assumed to be independent of the presence of alkyl substituents, but the

value of  $k_{add-O2}$  depends on both the degree and distribution of alkyl substituents, and is given by:

$$k_{\text{add-O2}} = k^{\circ}_{\text{add-O2}} \prod F_{i}(X), \text{ for } n = 0 \text{ (or 1)}$$
 (6)

$$k_{\text{add-O2}} = k^{\circ}_{\text{add-O2}} \prod F_{i}(X)/n^{0.5}, \text{ for } n \ge 1$$
 (7)

- Here, *n* is the number of alkyl substituents (in positions 1 to 5 relative to the addition of  $O_2$ ), and  $F_i(X)$  is the activating effect of each alkyl substituent in terms of its position (see Fig. 3). The assigned values of  $F_i(X)$  (given in Table 8) recreate the reported general trend in total hydroxyarene yields for methyl-substituted aromatics, and also a reasonable representation of the reported distribution of isomers formed from a given aromatic precursor (see Table S1). In the case of the toluene system, for example, the optimized parameters provide respective yields of 12.2 %, 3.7 % and 3.3 % for *o*-, *m*- and *p*-cresol, and a total rate coefficient of  $5.7 \times 10^{-16}$  cm<sup>3</sup> molecule<sup>-1</sup> s<sup>-1</sup> for the reaction of  $O_2$  with the set of OH-toluene adducts (i.e. HOC<sub>7</sub>H<sub>8</sub>) at 298 K; in very good
- agreement with the IUPAC recommendations (IUPAC, 2017c). To a first approximation, the simpler expression in Eq. (6) provides an acceptable description for the complete series of aromatics, but leads to a systematic underestimation of the hydroxyarene yields reported for *m*-xylene, *p*-xylene, 1,2,4-trimethylbenzene and 1,3,5-trimethylbenzene. The adjusted expression in Eq. (7) is therefore defined to allow a more precise description of the reported hydroxyarene yields for the more substituted species.
- As shown in Fig. 3, the two β-hydroxy cyclohexadienylperoxy radicals formed from O<sub>2</sub> addition are represented to undergo prompt ring closure to produce a common hydroxy-dioxa-bicyclo or "peroxide-bicyclic" radical. This process has been reported to dominate over alternative bimolecular reactions of the peroxy radicals under atmospheric conditions (e.g. Suh et al., 2003; Raoult et al., 2004; Glowacki et al., 2009; Wu et al., 2014; Li et al., 2014; Pan and Wang, 2014). The subsequent chemistry of the peroxide-bicyclic radical is shown in Fig. 4. In each case, the energy-rich radical either promptly isomerizes to form two cyclic
- epoxy-oxy radicals (as originally proposed by Bartolloti and Edney, 1995), or is stabilized and adds  $O_2$  to form two possible peroxide-bridged peroxy radicals. The cyclic epoxy-oxy radicals undergo ring-opening, followed by reaction with  $O_2$  to generate HO<sub>2</sub> and an epoxydicarbonylene product in each case. Evidence for the formation of the epoxydicarbonylene products has been

reported in experimental studies of a number of atmospheric systems (e.g. Yu and Jeffries, 1997; Kwok et al., 1997; Baltaretu et al., 2009; Birdsall et al., 2010; Birdsall and Elrod, 2011), although it is noted that their formation is calculated to be more important at reduced pressures (e.g. Glowacki et al., 2009; Li et al., 2014). In the present method, prompt isomerization of the peroxide-bicyclic radical to the cyclic epoxy-oxy radicals is assigned a total structure-independent branching ratio of 0.3, divided

- 5 equally between the two available routes. As indicated above, the subsequent chemistry leads to prompt formation of HO<sub>2</sub> (i.e. not delayed by first requiring conversion of an organic peroxy radical to an oxy radical via a bimolecular reaction), which supplements that formed in conjunction with the hydroxyarene (phenolic) products (see Fig. 3). Inclusion of the "epoxy-oxy" route with this optimized branching ratio results in total prompt HO<sub>2</sub> yields which provide a good representation of those reported by Nehr et al. (2011; 2012), and also total yields of the well-established  $\alpha$ -dicarbonyl products (formed from the alternative O<sub>2</sub>
- addition chemistry) that are consistent with those reported (see below). However, it is noted that this is an area of significant uncertainty, with theoretical studies predicting a much lower importance of the "epoxy-oxy" route at atmospheric pressure than applied here (e.g. Vereecken, 2018a; 2018b; and references therein). Further studies are required to elucidate the sources of epoxydicarbonylenes and prompt  $HO_2$  in aromatic systems.
- The (stabilized) peroxide-bicyclic radical possesses an allyl resonance, such that addition of O<sub>2</sub> can occur at two possible positions, as shown in Fig. 4. The overall rate coefficient is therefore given by  $k_{bc-add(1)} + k_{bc-add(2)}$ . The reference rate coefficient,  $k^{\circ}_{bc-add}$ , for a system with no alkyl substituents at either positions 'a' or 'b' (see Table 9) was assigned a value of  $4 \times 10^{-16}$  cm<sup>3</sup> molecule<sup>-1</sup> s<sup>-1</sup>, based on the total rate coefficient calculated for the peroxide-bicyclic radical formed in the benzene system by Glowacki et al. (2009). Reported calculations for methyl-substituted aromatics (e.g. Wu et al., 2014; Li et al., 2014) suggest that the value of  $k_{bc-add}$  for a given system is also potentially influenced by the presence of alkyl substituents in positions 'a' or 'b'. The addition rate coefficient estimated here is therefore given by:

$$k_{\text{bc-add}} = k^{\circ}_{\text{bc-add}} F_{a}(X) F_{b}(X)$$
(8)

Here,  $F_a(X)$  and  $F_b(X)$  quantify the activating effect of substituents in positions 'a' and 'b', respectively. The assigned values for an alkyl substituent (given in Table 9) allow a reasonable representation of the relative distribution of  $\alpha$ -dicarbonyl products (i.e. glyoxal, methylglyoxal and biacetyl) reported for the series of methyl-substituted aromatics for conditions when the peroxy

- 25 radicals react predominantly with NO (see Table S2). The large value of  $F_a(-alkyl)$  indicates that addition of  $O_2$  at an alkylsubstituted site in the resonant radical is strongly favoured, and can be assumed to be exclusive if only one of the two possible addition sites is alkyl substituted. The more modest influence of a substituent at position 'b' (characterized by  $F_b(-alkyl)$ ) also influences the relative formation of the specific  $\alpha$ -dicarbonyls (and their co-products) in cases where neither or both radical sites possess alkyl substituents. It is noted that the treatment of these structurally-complex allyl radicals differs from that reported in the 30 companion paper (Jenkin et al., 2018a) for generic allyl radicals, and is specific to this type of structure.
- The calculated yields presented in Table S2 also take account of minor formation of nitrate products from the peroxy + NO reactions (see Fig. 4), for which the currently estimated branching ratios vary from zero to 0.11 depending on peroxy radical structure. This is described in more detail elsewhere (Jenkin et al., 2018b). Table S2 also compares the calculated "prompt" yields

of HO<sub>2</sub> with those reported by Nehr et al. (2011; 2012) and the total nitrate yields with those reported by Rickard et al. (2010) and Elrod (2011). Fig. 5 presents a correlation plot of calculated and observed yields of hydroxyarenes (total and specific),  $\alpha$ dicarbonyls (total and specific), nitrates (total) and prompt HO<sub>2</sub>, which confirms that the methods presented above provide a reasonable representation of the first-generation OH-initiated chemistry of aromatic hydrocarbons. Sect. S6 provides example

calculations for the methods described above for the chemistry initiated by reaction of  $O_2$  with the OH-aromatic adducts formed from the addition of OH to toluene.

# 4.2 OH-aromatic oxygenate adducts

Product and mechanistic information on the reactions of adducts formed from the addition of OH radicals to aromatic oxygenates appears to be limited to those formed from hydroxyarene (phenolic) compounds (e.g. Olariu et al., 2002; Berndt et al., 2003;

- Coeur-Tourneur et al., 2006). Those studies have established that 1,2-dihydroxyarenes (catechols) and 1,4-benzoquinones are formed as ring-retaining products of the OH-initiated oxidation of phenol and cresols. On the basis of the reported information, the pathways presented in Fig. 6 are applied in relation to hydroxy-substituted aromatic compounds. Addition of OH at an unsubstituted carbon *ortho*- to an existing hydroxy substituent is assumed to result in exclusive formation of HO<sub>2</sub> and a 1,2dihydroxy product, following subsequent reaction of the adduct with O<sub>2</sub>. Addition of OH *para*- to an existing hydroxy substituent
- is assumed to result in formation of HO<sub>2</sub> and a reactive 4-hydroxy-cyclohexa-2,5-dienone product, following subsequent reaction of the adduct with O<sub>2</sub>. In cases where the initial addition of OH occurs at an unsubstituted carbon in the aromatic compound, further reaction of OH with the 4-hydroxy-cyclohexa-2,5-dienone partially produces a 1,4-benzoquinone product (as shown in Fig. 6), based on the methods applied to aliphatic compounds (Jenkin et al., 2018a).
- For other OH-aromatic oxygenate adducts, the mechanisms applied to OH-aromatic hydrocarbon adducts (see Sect. 4.1) are 20 provisionally applied, in the absence of information. Within the framework described in Sect. 4.1, some additional assumptions are applied in relation to addition of  $O_2$  to the (stabilized) resonant peroxide bicyclic radical, these being consistent with those applied generally to allyl radicals containing oxygenated substituents (Jenkin et al., 2018a). If the resonant peroxide bicyclic radical contains an oxygenated substituent at either or both positions 'a', addition of  $O_2$  is assumed to occur exclusively at the site possessing the substituent that is higher in the following list: -OH/-OR/-OOH/-OOR > -OC(=O)H/-OC(=O)R > alkyl/-H > -25 C(=O)H/-C(=O)R > -C(=O)OH/-C(=O)OR > -ONO<sub>2</sub> > -NO<sub>2</sub> (substituents with more remote oxygenated groups are treated as alkyl groups). If both sites possess a substituent of the same rating,  $O_2$  addition is assumed to occur equally at each site. An

#### **5** Reactions of organic radicals formed from OH attack on substituent groups

oxygenated substituent at position 'b' is assumed to have no effect (Table 9).

Carbon-centred organic radicals (R) formed from H-atom abstraction from, or OH addition to, substituent groups in aromatic compounds generally react as described for those formed from aliphatic organic compounds in the companion paper (Jenkin et al., 2018a). In the majority of cases, therefore, they react rapidly and exclusively with molecular oxygen (O<sub>2</sub>) under tropospheric conditions, to form the corresponding thermalized peroxy radicals ( $RO_2$ ), the chemistry of which will be summarized elsewhere (Jenkin et al., 2018b):

$$R + O_2 (+M) \rightarrow RO_2 (+M) \tag{R1}$$

(M denotes a third body, most commonly, N<sub>2</sub>). Abstraction of hydrogen from hydroxy and hydroperoxy substituents groups 5 in aromatic VOCs results in formation of phenoxy and phenyl peroxy radicals, respectively. The representation of phenyl peroxy radical chemistry will be considered elsewhere, along with that of other peroxy radicals (Jenkin et al., 2018b). The chemistry of phenoxy radicals differs from that of oxy radicals in general, in that they apparently do not react with  $O_2$ , isomerize or decompose under tropospheric conditions. Kinetics studies for the phenoxy radical itself ( $C_6H_5O$ ) indicate that reactions with NO, NO<sub>2</sub> and O<sub>3</sub> are likely to be competitive under ambient conditions (Platz et al., 1998; Berho et al., 1998;

10 Tao and Li, 1999), with evidence also reported for reaction with HO<sub>2</sub> at room temperature (Jenkin et al., 2007; 2010) and in low temperature combustion systems (Herbinet et al., 2013). As summarized in Table 10, the reactions with NO<sub>2</sub>, O<sub>3</sub> and HO<sub>2</sub> are generally represented for a given phenoxy radical, although reaction with NO<sub>2</sub> is unavailable for phenoxy radicals with two *ortho*- substituents, because formation of a 1-hydroxy-2-nitroarene product is precluded. The reaction with NO is not represented because the reverse reaction is reported to occur on the timescale of about one minute (Berho et al., 1998).

#### 15 6 Conclusions

A structure activity relationship (SAR) method has been developed to estimate rate coefficients for the reactions of the OH radical with aromatic organic species. This group contribution method was optimized using a database including a set preferred rate coefficients for 67 species. The overall performance of the SAR in determining log  $k_{298K}$  is now summarized. The distribution of errors (log  $k_{calc}/k_{obs}$ ), the Root Mean Squared Error (RMSE), the Mean Absolute Error (MAE) and the

20 Mean Bias Error (MBE) were examined to assess the overall reliability of the SAR. The RMSE, MAE and MBE are here defined as:

$$RMSE = \sqrt{\frac{1}{n} \sum_{i=1}^{n} (\log k_{calc} - \log k_{obs})^2}$$
(9)

$$MAE = \frac{1}{n} \sum_{i=1}^{n} \left| \log k_{calc} - \log k_{obs} \right| \tag{10}$$

$$MBE = \frac{1}{n} \sum_{i=1}^{n} (\log k_{calc} - \log k_{obs})$$
<sup>(11)</sup>

25 where *n* is the number of species in the dataset. The assessment was performed to identify possible biases within a series of categories, namely hydrocarbons, monofunctional oxygenated species and bifunctional oxygenated species. Errors computed for these subsets are summarized Fig. 7, where they are compared with those for the corresponding categories of aliphatic organic compound, as reported in the companion paper (Jenkin et al., 2018a).

The calculated log  $k_{298K}$  shows no significant bias, with MBE remaining below 0.02 log units for the various subsets, and with median values of the error distributions close to zero. The reliability of the SAR decreases with the number of oxygenated functional groups on the aromatic ring, with the RMSE increasing from 0.06 for hydrocarbons to 0.07 for monofunctional and 0.08 for bifunctional species, i.e. a relative error for the calculated  $k_{298K}$  of a 15 %, 17 % and 20 %,

- respectively. This shows a similar pattern to that reported previously for the much larger dataset of aliphatic species (Jenkin et al., 2018a), but with systematically lower errors. As described in Sect. 3.2, some of the classes of aromatic oxygenated species contain data for only a single compound, such that the optimized parameters inevitably provide a good description of the observed data; whereas the aliphatic data are typically comprised of larger and more diverse sets of species. Additional rate coefficients would therefore be highly valuable for further assessment and evaluation of the SAR for a variety of aromatic
- oxygenated species. Finally, for the full database, the SAR gives generally reliable  $k_{298K}$  estimates, with a MAE of 0.04 and a RMSE of 0.07, corresponding to an overall agreement of the calculated  $k_{298K}$  within 17%.

#### Acknowledgements

This work received funding from the Alliance of Automobile Manufacturers, and as part of the MAGNIFY project, with funding from the French National Research Agency (ANR) under project ANR-14-CE01-0010, and the UK Natural

Environment Research Council (NERC) via grant NE/M013448/1. It was also partially funded by the UK National Centre for Atmospheric Sciences (NCAS) Composition Directorate. Marie Camredon (LISA, Paris) and Mike Newland (University of York) are gratefully acknowledged for helpful discussions on this work. We also thank Luc Vereecken (Forschungszentrum Jülich) for providing detailed comments during the open discussion, and two anonymous referees for review comments, that helped to improve the manuscript.

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

 IUPAC
 Task
 Group
 on
 Atmospheric
 Chemical
 Kinetic
 Data
 Evaluation.:
 <u>http://iupac.pole-</u>

 ether.fr/htdocs/datasheets/pdf/HOx\_AROM2\_HO\_toluene.pdf, 2017a.
 2017a.
 2017a.
 2017a.

IUPAC Task Group on Atmospheric Chemical Kinetic Data Evaluation.: <u>http://iupac.pole-</u> ether.fr/htdocs/datasheets/pdf/AROM\_RAD1\_HOC6H6\_O2.pdf, 2017b. IUPAC Task Group on Atmospheric Chemical Kinetic Data Evaluation.: <u>http://iupac.pole-</u> ether.fr/htdocs/datasheets/pdf/AROM RAD4 HOC7H8 O2.pdf, 2017c.

Jenkin, M. E. and Clemitshaw, K. C.: Ozone and other secondary photochemical pollutants: chemical processes governing their formation in the planetary boundary layer, Atmos. Environ., 34, 2499–2527, 2000.

5 Jenkin, M. E., Saunders, S. M., Wagner, V. and Pilling, M. J.: Protocol for the development of the Master Chemical Mechanism, MCM v3 (Part B): tropospheric degradation of aromatic volatile organic compounds. Atmos. Chem. Phys. 3, 181-193, 2003.

Jenkin, M. E., Hurley, M. D., and Wallington, T. J.: Investigation of the radical product channel of the  $CH_3C(O)O_2 + HO_2$  reaction in the gas phase, Phys. Chem. Chem. Phys., 9, 3149-3162, 2007.

- Jenkin, M. E., Hurley, M. D., and Wallington, T. J.: Investigation of the radical product channel of the CH<sub>3</sub>OCH<sub>2</sub>O<sub>2</sub> + HO<sub>2</sub> reaction in the gas phase, J. Phys. Chem. A, 114, 408-416, 2010.
   Jenkin, M. E., Derwent, R. G., and Wallington, T. J.: Photochemical ozone creation potentials for volatile organic compounds: Rationalization and estimation, Atmos. Environ., 163, 128-137, 2017.
   Jenkin, M. E., Valorso, R., Aumont, B., Rickard, A. R. and Wallington, T. J.: Estimation of rate coefficients and branching
- 15 ratios for reactions of OH with aliphatic organic compounds for use in automated mechanism construction, Atmos. Chem. Phys. Discuss., https://doi.org/10.5194/acp-2018-145, in review, 2018a. Jenkin et al.: Estimation of rate coefficients and branching ratios for reactions of organic peroxy radicals for use in

automated mechanism construction, in preparation, 2018b.

Klotz, B., Sørensen, S., Barnes, I., Becker, K.H., Etzkorn, T., Volkamer, R., Platt, U., Wirtz, K. and Martín-Reviejo, M.:

20 Atmospheric oxidation of toluene in a large volume outdoor photoreactor: in situ determination of ring-retaining product yields, J. Phys. Chem. A, 102, 10289-10299, 1998.

Kwok, E.S.C. and Atkinson R.: Estimation of hydroxyl radical reaction rate constants for gas-phase organic compounds using a structure-reactivity relationship: an update, Atmos. Environ., 29, 1685–1695, 1995.

Kwok, E. S. C., Aschmann, S. M., Atkinson, R. and Arey, J.: Products of the gas-phase reactions of o-, m- and p-xylene with the OH radical in the presence and absence of NO<sub>x</sub>, J. Chem. Soc., Faraday Trans., 93(16), 2847-2854, 1997.

Lewis, A. C., Evans, M. J., Hopkins, J. R., Punjabi, S., Read, K. A., Purvis, R. M., Andrews, S. J., Moller, S. J., Carpenter, L. J., Lee, J. D., Rickard, A. R., Palmer, P. I., and Parrington, M.: The influence of biomass burning on the global distribution of selected non-methane organic compounds, Atmos. Chem. Phys., 13, 851-867, https://doi.org/10.5194/acp-13-851-2013, 2013.

Li, Y. and Wang, Y.: The atmospheric oxidation mechanism of 1,2,4-trimethylbenzene initiated by OH radicals, Phys. Chem. 30 Chem. Phys., 16, 17908-17917, 2014.

Loison, J-C., Rayez, M-T., Rayez, J-T., Gratien, A., Morajkar, P., Fittschen, P. and Villenave, E.: Gas-phase reaction of hydroxyl radical with hexamethylbenzene. J. Phys. Chem. A, 116, 12189–12197, 2012.
Maleknia, S. D., Bell, T. L. and Adams, M. A.: PTR-MS analysis of reference and plant-emitted volatile organic compounds, Int.

J. Mass. Spectrom., 262, 203-210, 2007.

Mehta, D., Nguyen, A., Montenegro, A. and Li, Z.: A kinetic study of the reaction of OH with xylenes using the relative rate/discharge flow/mass spectrometry technique, J. Phys. Chem. A, 113, 12942–12951, 2009.

Misztal, P. K., Owen, S. M., Guenther, A. B., Rasmussen, R., Geron, C., Harley, P., Phillips, G. J., Ryan, A., Edwards, D. P., Hewitt, C. N., Nemitz, E., Siong, J., Heal, M. R., and Cape, J. N.: Large estragole fluxes from oil palms in Borneo, Atmos. Chem. Phys., 10, 4343-4358, https://doi.org/10.5194/acp-10-4343-2010, 2010.

Misztal, P. K., Hewitt, C. N., Wildt, J., Blande, J. D., Eller, A. S. D., Fares, S., Gentner, D. R., Gilman, J. B., Graus, M., Greenberg, J., Guenther, A. B., Hansel, A., Harley, P., Huang, M., Jardine, K., Karl, T., Kaser, L., Keutsch, F. N., Kiendler-Scharr, A., Kleist, E., Lerner, B. M., Li, T., Mak, J., Nölscher, A. C., Schnitzhofer, R., Sinha, V., Thornton, B., Warneke, C., Wegener, F., Werner, C., Williams, J., Worton, D. R., Yassaa, N. and Goldstein, A. H.: Atmospheric benzenoid emissions

5

- from plants rival those from fossil fuels, Scientific Reports, 5, Article number 12064, doi:10.1038/srep12064, 2015.
   Mousavipour, S. H. and Homayoon, Z.:Multichannel RRKM-TST and CVT rate constant calculations for reactions of CH<sub>2</sub>OH or CH<sub>3</sub>O with HO<sub>2</sub>, J. Phys. Chem. A, 115, 3291-3300, 2011.
   Nehr, S, Bohn, B., Fuchs, H., Hofzumahaus, A. and Wahner, A: HO<sub>2</sub> formation from the OH + benzene reaction in the presence of O<sub>2</sub>, Phys. Chem. Chem. Phys., 13, 10699–10708, 2011.
- 15 Nehr, S, Bohn, B. and Wahner, A: Prompt HO<sub>2</sub> formation following the reaction of OH with aromatic compounds under atmospheric conditions, J. Phys. Chem. A, 116, 6015-6026, 2012. Nishino, N., Arey, J. and Atkinson, R: Formation yields of glyoxal and methylglyoxal from the gas-phase OH radical-initiated reactions of toluene, xylenes, and trimethylbenzenes as a function of NO<sub>2</sub> concentration, J. Phys. Chem. A, 114, 10140–10147, 2010.
- Noda, J., Volkamer, R., and Molina, M.J.: Dealkylation of alkylbenzenes: a significant pathway in the toluene, o-, m-, and p-xylene + OH reaction, J. Phys. Chem. A., 113, 9658-9666, 2009.
  Odum, J. R., Jungkamp, T. P. W., Griffin, R. J., Forstner, H. J. L., Flagan, R. C. and Seinfeld, J. H.: Aromatics, reformulated gasoline, and atmospheric organic aerosol formation. Environ. Sci. Technol., 31 (7), 1890-1897, 1997.
  Olariu, R. I., Klotz, B., Barnes, I., Becker, K. H., and Mocanu, R.: FT-IR study of the ring-retaining products from the reaction of
- OH radicals with phenol, o-, m-, and p-cresol, Atmos. Environ., 36 (22), 3685-3697, 2002.
   Owen, S. M., Boissard, C., and Hewitt, C. N.: Volatile organic compounds (VOCs) emitted from 40 Mediterranean plant species: VOC speciation and extrapolation to habitat scale, Atmos. Environ., 35, 5393-5409, 2001.
   Pan, S. and Wang, L.: Atmospheric oxidation mechanism of m-xylene initiated by OH radical, J. Phys. Chem. A, 118, 10778-10787, 2014.
- 30 Passant, N. R.: Speciation of UK emissions of non-methane volatile organic compounds. AEA Technology Report ENV-0545. Culham, Abingdon, United Kingdom, 2002. Peeters, J., Vandenberk, S., Piessens, E. and Pultau, V.: H-atom abstraction in reactions of cyclic polyalkenes with OH,

Chemosphere, 38, 1189-1195, 1999.

Peeters, J., Boullart, W., Pultau, V., Vandenberk, S. and Vereecken, L.: Structure-activity relationship for the addition of OH to (poly)alkenes: site-specific and total rate constants, J. Phys. Chem. A, 111, 1618-1631, 2007.

Perry, R. A., Atkinson, R. and Pitts, J. N.: Kinetics and mechanism of the gas phase reaction of OH radicals with methoxybenzene and o-cresol over the temperature range 299- 435 K, J. Phys. Chem., 81, 1607-1611, 1977.

- 5 Platz, J., Nielsen, O. J., Wallington, T. J., Ball, J. C., Hurley, M. D., Straccia, A. M., Schneider, W. F., and Sehested, J.: Atmospheric chemistry of the phenoxy radical, J. Phys. Chem. A, 102, 7964-7974, 1998.
  Raoult, S, Rayez, M-. T., Rayez, J-. C. and Lesclaux, R.: Gas phase oxidation of benzene: Kinetics, thermochemistry and mechanism of initial steps, Phys. Chem. Chem. Phys., 6, 2245-2253, 2004.
  Rickard, A. R., Wyche, K. P., Metzger, A., Monks, P. S., Ellis, A. M., Dommen, J., Baltensperger, U., Jenkin, M. E. and Pilling
- 10 M. J.: Gas phase precursors to anthropogenic secondary organic aerosol: using the Master Chemical Mechanism to probe detailed observations of 1,3,5-trimethylbenzene photo-oxidation, Atmos. Environ., 44, 5423-5433, 2010. Saunders, S. M., Jenkin, M. E., Derwent, R. G. and Pilling, M. J.: Protocol for the development of the Master Chemical Mechanism, MCM v3 (Part A): tropospheric degradation of non-aromatic volatile organic compounds, Atmos. Chem. Phys., 3, 161-180, 2003.
- 15 Smith, D. F., McIver, C. D., and Kleindienst, T. E.: Primary product distribution from the reaction of hydroxyl radicals with toluene at ppb NO<sub>x</sub> mixing ratios, J. Atmos. Chem. 30, 209–228, 1998.
  Smith, D. F., Kleindienst, T. E., and McIver, C. D.: Primary product distributions from the reaction of OH with m-, p-xylene,

1,2,4- and 1,3,5-trimethyl benzene, J. Atmos. Chem., 34, 339-364, 1999.

Suh, I., Zhang, D., Zhang, R., Molina, L. T. and Molina, M. J.: Theoretical study of OH addition reaction to toluene. Chem. Phys. 20 Lett., 364, 454–462, 2002.

Suh, I., Zhang, R., Molina, L. T. and Molina, M. J.: Oxidation mechanism of aromatic peroxy and bicyclic radicals from OHtoluene reactions, J. Am. Chem. Soc., 125, 12655–12665, 2003.

Tao, Z. and Li, Z.: A kinetics study on reactions of  $C_6H_5O$  with  $C_6H_5O$  and  $O_3$  at 298K, Int. J. Chem. Kinetics, 31, 65-72, 1999.

- 25 Tuazon, E., Arey, J., Atkinson, R. and Aschmann, S.: Gas-phase reactions of vinylpyridine and styrene with OH and NO<sub>3</sub> radicals and O<sub>3</sub>, Environ. Sci. Technol., 27, 1832-1841, 1993. Ulman M. et al.: Determination of volatile organic compounds (VOCs) in the atmosphere over central Siberian forest and southern part of European Taiga in Russia, Chemia Analityczna, 52, 435-451, 2007. Vereecken, L.: Reaction mechanisms for the atmospheric oxidation of monocyclic aromatic compounds, in Advances in
- 30 Atmospheric Chemistry, Vol. 2, Barker, J. R., Allison, S., Wallington, T. J. (Ed.), Accepted for publication, 2018a. Vereecken, L.: Interactive comment on "Estimation of rate coefficients and branching ratios for gas-phase reactions of OH with aromatic organic compounds for use in automated mechanism construction" by Michael E. Jenkin et al., Atmos. Chem. Phys. Discuss., https://doi.org/10.5194/acp-2018-146-SC1, 2018b.

Volkamer, R., Platt, U., and Wirtz, K.: Primary and secondary glyoxal formation from aromatics: experimental evidence for the bicycloalkyl-radical pathway from benzene, toluene and p-xylene, J. Phys. Chem. A, 105, 7865–7874, 2001.
Volkamer, R., Klotz, B., Barnes, I., Imamura, T., Wirtz, K., Washida, N., Becker, K. H., and Platt, U.: OH-initiated oxidation of benzene – Part I. Phenol formation under atmospheric conditions, Phys. Chem. Chem. Phys., 4 (9), 1598–1610, 2002.

5 Went, F. W.: Blue hazes in the atmosphere, Nature, 187, 641–645, 1960.
Wu, R., Pan, S., Li, Y. and Wang., L.: Atmospheric oxidation mechanism of toluene. J. Phys. Chem. A, 118, 4533–4547, 2014.
Yu, J. Z. and Jeffries, H. E.: Atmospheric photooxidation of alkylbenzenes 2. Evidence for formation of epoxide intermediates, Atmos. Environ., 31 (15), 2281–2287, 1997.