# Peer review of "Estimation of rate coefficients and branching ratios for gas-phase reactions of OH with aromatic organic compounds for use in automated mechanism construction"

_Atmospheric Chemistry and Physics, 2018_

## Referee Comment (RC1) · Anonymous Referee #1 · 15 Mar 2018

I applaud the authors for tackling this messy and complex problem. This is a valuable paper and should be published; ACP is an appropriate journal for this manuscript. The authors have laid out the case well, described the methodology in great detail, and been transparent about assumptions. While the lack of data for some configurations makes it difficult to develop robust generalizations for similar structures, especially in product distribution, it is valuable to have a protocol. The authors present one that is vastly improved over what we have now (basically nothing generalizable). I hope that the community assists in improving this by collecting more data to help extend and

evaluate the subsequent reaction pathways of the less-studied oxygenated aromatics and their products.

A valuable component of this paper is the development of site-specific attack distribution and prediction of resulting product structures. This is necessary for so many issues – yield of ozone per molecule of VOC, yield of SOA (and developing a mechanistic aerosol mechanism), source attribution based on secondary products, etc. The focus is on automated detailed mechanisms but it would also be useful for people trying to write balanced chemical equations for individual chemicals. It's surprising how good the yields are in Figure 5.

The illustration of calculating SARs for several molecules is valuable; it should be referenced in the main paper somewhere - I cannot find a mention of it. The method for aromatics is different enough than previously applied for simpler molecules, including the additional correction factors (i.e. the $exp(140/T)$ for additional methyl groups; substituent adjustment factors; the use of alternate k values), thus directing readers to the end of SI would help make it clearer (versus them finding their own way to the end of it). In the examples, it would be useful to explicitly list where R=1 or F=1, for example: in the (b) carbon of p-cymene, I'm assuming it should be ktert*F(CH3)*F(CH3)*F(Ph2), where F(CH3)=1 so it is not shown? Also would be informative to see an example of the calculations for addition reactions of O2 to OH-aromatic adducts (I didn't calculate the same distribution of cresols as reported – likely misinterpreting how substituent factors are applied in this case).

For phenols and cresols, the authors recommend using experimental data when available. Are there other configurations where experimental data should override the estimates?

Below, are some specific comments:

Page 4, line 1: would be helpful to kprim, ksec and ktert so that reader does not have to search through another paper – could add to Table 1, or list in text. Perhaps add

[Figure]

kabs(OH). The 2018a paper is a critical companion paper, but this one should also stand mainly on its own.

Table 1: It took me a while to figure out that "substituent" is not the neighboring group (i.e. not the "X" in Kwok and Atkinson tables), but the successive carbons in the alkyl group, and the "X" is the aromatic ring. Adding F(-CH3) and other groups might help – or state that readers can find these other Fs in the 2018a paper.

Table 3: If the OH addition is on an ipso carbon of a compound with 3 substituents, I assume one uses the substituent factor for just the other 2 substituents (i.e. number of substituents = 2). Might state that in the paper. Title of Table 3 reads "Each factor relates to the combination of methyl substitutions", but it also relates to other functional groups.

Page 4, line 20: I don't know if H-abstraction is "minor", if you later present it (Table 4) as 3-22%.

Page 5, equation 4: shouldn't this be kadd*F(phi)*R(phi)? R is not needed until later, and not introduced until Table 6, but Table 6 does include R for methyl=1, so this would better generalize the equation.

Page 8, line 5: So you totally ignore the aromatic carbons and use the estimated rate for the alkenyl group?

. . .and some technical corrections/comments:

Page 5, line 18: Can't see that you defined kcalc , assume it is the same as k, defined as k=kadd+kabs (page 3, line 17)

Consider replacing the "." in equations with "·" or "x" to signify multiplication. It looks like a period.

Page 8, line 2: replace reference to Table 3 with Table 6.

Page 9, line 8: replace "upper panel" with "large panel" or "main panel".

Table 8: check the title. Page 12, line 10: replace reference to Fig. 4 with Fig. 3

[Figure]

---

## Referee Comment (RC2) · Anonymous Referee #2 · 21 Mar 2018

General comments

This manuscript describes the development of a structure-activity relationship (SAR) model for the reactions of OH with aromatic organic compounds which is explicit in terms of the OH reaction mechanism, and in subsequent reactions with molecular oxygen. These features likely will assist in the development of more detailed and quantitively correct representations of the atmospheric oxidation mechanisms for aromatic compounds. The work is carefully planned and performed, and the topical nature of the work makes it quite appropriate for publication in the Atmospheric Chemistry and

[Figure]

Physics. The supporting information is very through and includes all experimental and calculated rate constant values, as well as examples to help the reader calculate rate constants from the SAR parameters.

Specific comments

p.2: It would be good to note as a motivation that there really isn't a generalizable OH + aromatic SAR currently available in the literature.

p. 5, line 11: I assume from this discussion that the regression didn't use the experimental uncertainties in the rate constants to weight the individual values. Was anything done to take into account that the experimental rate constants have varying uncertainties?

p. 5, line 13: The comment about ortho- and para-substituents being more activating than meta-substituents is only true for the specific case of electron donating substituents such as methyl groups, which is also a well-known property of electrophilic aromatic substitution reactions.

Various tables: Why aren't uncertainties given for the various F(phi) values determined from the fitting process?

p. 6 line 16: I don't understand the problem being described here. From the statement earlier in this paragraph, I thought the H-abstraction values were being determined from p-cymene (the only compound for which H-abstraction experimental information is available), so I don't understand why these parameters then need to be adjusted. Additionally, on what theoretical grounds might these adjustments be justified?

p. 6 line 30: The equation for R(phi) should be explicitly given.

Table 5: I understand that previous reports used different definitions for the branching ratio, but it is quite distracting and confusing to have two sets of values reported. I suggest that the authors convert all branching ratios to a common definition and to report that single set of values.

The form of equation 5 should be justified in the text.

Table S3: The authors should use the term "calculated" rather than "estimated" to be consistent with the other instances where rate constants calculated from the SAR model are reported. I understand that the red font entries in the "recommended" column are experimental values, but for the non-red font entries, what is the process for the determination of these recommended values?

Figure 7: To what extent is the lower uncertainty evident in the aromatic set of compounds as compared to the aliphatic compounds a function of larger structural/functional group heterogeneity of the aliphatic compound group? Or is there another explanation?

Technical corrections

Equation 4: The product sign between the two terms in the summation argument looks more like a decimal point. I suggest removing it entirely.

---

## Short Comment (SC1) · 16 Apr 2018

**Comment** on "*Estimation of rate coefficients and branching ratios for gas-phase reactions of OH with aromatic organic compounds for use in automated mechanism construction*" by Jenkin et al., ACPD, 2018

**Luc Vereecken**

This is an excellent paper, summarizing many aspects of aromatic chemistry. I will not comment on most of it, as I generally agree with how the analysis is done. Aromatic chemistry is very complex, and the authors are the first to propose useable SARs that capture rate coefficient and site-specificity trends.

There is, however, one aspect I feel is not in agreement with the experimental observations and the theoretical data available. In particular, I have reservations on a mechanism that incorporates the chemically activated "peroxide-bicyclic" radical (**BCP-yl** in figure LV-1) as an important source of up to 30% of the products. My reasoning is described below in several parts; it is obviously based on incomplete data and thus by no means final. Many of my remarks are based on my recent overview of theoretical studies on aromatic chemistry, of which the authors have received a copy (though much too late to incorporate that data in the paper discussed here). I have summarized the chemical mechanism in figure LV-1 below, and will use the naming and reaction labels in that figure.

I recognize that the proposed mechanism (Figure 3 and 4 in Jenkin et al. 2018) is only a small part of the paper, and this comment is thus not a major criticism of that work. My worry is mainly that no systematic mechanistic improvement can be done by updating rate coefficients and yields of elementary reactions, if the mechanism in the model does not match the underlying chemical process. Given that this mechanism would be implemented in the MCM, the most commonly used semi-explicit mechanism, this could hamper progress significantly. This comment is not as complete as I wanted it to be and may contain errors, as I ran out of time trying to meet the deadline for comment submission. I apologize for the poor presentation of this text and the lack of a more thorough numerical analysis, and remain available to clarify this text.

---

## Author Comment (AC1) · 28 May 2018

Authors' responses to referee and discussion comments on: Jenkin et al., Atmos. Chem. Phys. Discuss., https://doi.org/10.5194/acp-2018-146, 2018.

We are very grateful to the referees and commenter for their supportive comments on this work, and for their helpful suggestions for modifications and improvements. Responses to the comments are now provided (the original comments are shown in blue font).

**A. Comments by Referee 1**

**Opening comments:**

I applaud the authors for tackling this messy and complex problem. This is a valuable paper and should be published; ACP is an appropriate journal for this manuscript. The authors have laid out the case well, described the methodology in great detail, and been transparent about assumptions. While the lack of data for some configurations makes it difficult to develop robust generalizations for similar structures, especially in product distribution, it is valuable to have a protocol. The authors present one that is vastly improved over what we have now (basically nothing generalizable). I hope that the community assists in improving this by collecting more data to help extend and evaluate the subsequent reaction pathways of the less-studied oxygenated aromatics and their products.

A valuable component of this paper is the development of site-specific attack distribution and prediction of resulting product structures. This is necessary for so many issues – yield of ozone per molecule of VOC, yield of SOA (and developing a mechanistic aerosol mechanism), source attribution based on secondary products, etc. The focus is on automated detailed mechanisms but it would also be useful for people trying to write balanced chemical equations for individual chemicals. It's surprising how good the yields are in Figure 5.

Response: We are very grateful to the referee for these very positive and supportive comments on our work.

**Comment A1**: The illustration of calculating SARs for several molecules is valuable; it should be referenced in the main paper somewhere - I cannot find a mention of it. The method for aromatics is different enough than previously applied for simpler molecules, including the additional correction factors (i.e. the exp(140/T) for additional methyl groups; substituent adjustment factors; the use of alternate k values), thus directing readers to the end of SI would help make it clearer (versus them finding their own way to the end of it). In the examples, it would be useful to explicitly list where R=1 or F=1, for example: in the (b) carbon of p-cymene, I'm assuming it should be ktert\*F(CH3)\*F(CH3)\*F(Ph2), where F(CH3)=1 so it is not shown? Also would be informative to see an example of the calculations for addition reactions of O2 to OH-aromatic adducts (I didn't calculate the same distribution of cresols as reported – likely misinterpreting how substituent factors are applied in this case).

Response: We agree that clearer reference to the example SAR calculations would be helpful. Reference was originally made in Sect. 3.1.1 to the calculations for methyl-substituted aromatic hydrocarbons in the Supplement. Similarly to that in the preceding companion paper, the following statement has now also been added to the description of the scope of the paper in the Introduction (new text in red font):

"..... In each case, the rate coefficient is defined in terms of a summation of partial rate coefficients for H atom abstraction or OH addition at each relevant site in the given organic compound, so that the attack distribution is also defined. This is therefore the first generalizable SAR for reactions of OH with aromatic compounds that aims to capture observed trends in rate coefficients and the site-specificity of attack. Application of the methods is illustrated with examples in the Supplement."

Although the example SAR calculations are located towards the end of the Supplement, they are also clearly advertised in the contents list on page 1 of the Supplement.

The referee makes a valid point about the omission of the unity  $F(-CH_3)$  factor in the example OH + aromatic calculations, and this has been included in the revised Supplement to improve clarity. We also agree with the referee concerning the inclusion of example calculations for the subsequent reaction sequences initiated by reaction with O2. Full illustration of the complete reaction sequences

following the reaction of OH with toluene has now been included in the Supplement, with reference to this at the end of Sect. 4.1, as follows:

"Sect. S6 provides example calculations for the methods described above for the chemistry initiated by reaction of  $O_2$  with the OH-aromatic adducts formed from the addition of OH to toluene."

**Comment A2**: For phenols and cresols, the authors recommend using experimental data when available. Are there other configurations where experimental data should override the estimates?

Response: Estimation methods are generally used to fill in (the often large) gaps in knowledge, where experimental data is unavailable. As a general rule, therefore, a recommended parameter based on evaluated experimental data for a specific reaction should always override an estimated parameter – even if methods based on SARs are initially used to construct a highly detailed mechanism for efficiency. A statement to this effect is actually included in Sect. 3.2.1, where phenol and the cresols are discussed. In the present work, the estimated parameters were generally found to be very close to the recommendations. Although the estimated rate coefficients and attack distributions for phenol and the cresols recreate some of the features inferred from reported experimental studies, we judged that the deviations were sufficient that it was helpful to emphasise this point and also to provide recommended attack distributions based on experimental information.

**Below, are some specific comments:**

**Comment A3**: Page 4, line 1: would be helpful to kprim, ksec and ktert so that reader does not have to search through another paper – could add to Table 1, or list in text. Perhaps add kabs(OH). The 2018a paper is a critical companion paper, but this one should also stand mainly on its own.

Response: The paragraph the referee is referring to provides an overview of the relevant information for saturated organic compounds that can be found in the preceding companion paper. In addition to  $k_{\text{prim}}$ ,  $k_{\text{sec}}$  and  $k_{\text{tert}}$ , this includes a (potentially large) number of neighbouring group parameters, F(X), and a series of generic rate coefficients for reactions at oxygenated groups, including  $k_{\text{abs(-OH)}}$  (as summarised in the relevant paragraph). We judged that reproduction of all this information could not be justified, and that it would be artificial and misleading to reproduce a subset of it in the present paper (e.g. only those parameters mentioned in the first line of the paragraph, as suggested by the referee). In some respects, the companion paper serves as fully-referenced supporting information, and we feel this is sufficient and appropriate.

**Comment A4**: Table 1: It took me a while to figure out that "substituent" is not the neighboring group (i.e. not the "X" in Kwok and Atkinson tables), but the successive carbons in the alkyl group, and the "X" is the aromatic ring. Adding F(-CH3) and other groups might help – or state that readers can find these other Fs in the 2018a paper.

Response: The use of the term "substituent" for both the substituent group on the aromatic ring and generally for the neighbouring group in the Kwok and Atkinson method is unfortunate. It is difficult to see a way to change this, because both are common and valid uses of the term.

For clarification here, the parameters F(-Ph1) and F(-Ph2) in the present work are completely analogous to the parameter (or substituent factor)  $F(-C_6H_5)$  in Table 2 of Kwok and Atkinson (1995), and apply when H atom abstraction is occurring from the carbon atom adjacent to the aromatic ring (i.e. in a substituent to the aromatic ring). In the present work, we found it was necessary to define the two parameters, F(-Ph1) and F(-Ph2), with each applying to a different set of substituents to the aromatic ring. Thus, column 1 identifies the relevant substituent to the aromatic ring from which abstraction is occurring; and column 2 gives the relevant parameter (or neighbouring group substituent factor) that is applied in each case to account for the neighbouring aromatic group effect. Although at first sight this is possibly a little confusing, we feel that the context in the associated text and the information given in the comments to the table should clarify the approach.

In view of the referee's comment, we have slightly changed the table caption to remove the double use of the term "substituent". This now reads:

"Neighbouring group factors, F(X), for  $\alpha$ - H-atom abstraction from substituents in aromatics, and their temperature dependences described by F(X) = AF(X) exp(-BF(X)/T)."

**Comment A5**: Table 3: If the OH addition is on an ipso carbon of a compound with 3 substituents, I assume one uses the substituent factor for just the other 2 substituents (i.e. number of substituents = 2). Might state that in the paper. Title of Table 3 reads "Each factor relates to the combination of methyl substitutions", but it also relates to other functional groups.

Response: We are pleased that the referee has understood the method correctly. We believe the point being made is already clearly stated in Sect. 3.1.1, just after Eq. (4), where the following text appears:

".....where k is either  $k_{arom}$  or  $k_{ipso}$  and  $F(\Phi)$  is a factor that accounts for the effect of the combination of methyl substituents in the molecule in terms of their positions (i.e. *ortho-*, *meta-* or *para-*) relative to each OH addition location."

The omission of an effect of the *ipso*- substituent within  $F(\Phi)$  is logical, because the product radical is delocalized over the carbon atoms *ortho-*, *meta-* and *para-* to the OH addition location (e.g. see Fig. 3).

The same statement about  $F(\Phi)$  is also made in the caption to Table 3, where the  $F(\Phi)$  values are presented. These values are specific to the combination of methyl substituents. The referee is correct that these values are also used (modified by the adjustment factors,  $R(\Phi)$ , in Table 6) for other substituents. We feel this is already very clearly explained, and that it would be confusing to state that the parameters in Table 3 also apply to other substituents.

**Comment A6**: Page 4, line 20: I don't know if H-abstraction is "minor", if you later present it (Table 4) as 3-22%.

Response: The text being referred to on page 4 concerns *methyl-substituted aromatics* (Sect. 3.1.1) for which the contributions of H abstraction are reported to vary from 4 % for *m*-xylene to 13.7 % for hexamethylbenzene (Table 4), these generally being reported as "minor contributions" in the cited studies. We feel that the calculated H atom abstraction contribution for *p*-cymene of 22 %, quoted by the referee, probably also classifies as minor. However, this is discussed in the subsequent section on *higher alkyl-substituted aromatics* (Sect. 3.1.2) and is not encompassed by the statement on page 4 and therefore not relevant. Clearly if a substituent is large enough, H abstraction from that substituent can make a major contribution.

**Comment A7**: Page 5, equation 4: shouldn't this be kadd\*F(phi)\*R(phi)? R is not needed until later, and not introduced until Table 6, but Table 6 does include R for methyl=1, so this would better generalize the equation.

Response: The referee raises a valid point. However, we feel it is clearer to delay introducing and discussing the adjustment factors,  $R(\Phi)$ , until they are required in later sections. The unity value presented for CH3 in Table 6 is included for completeness, and emphasizes that it is a reference case. In view of the referee's comment (and comment B7 of referee 2) we have now formalised the method as suggested, with inclusion of  $R(\Phi)$  in Sect. 3.1.2 on *higher alkyl-substituted aromatics*, as follows:

"Table 6 shows a set of adjustment factors for non-methyl substituents,  $R(\Phi)$ , that represent corrections to the values of  $F(\Phi)$  in Table 3 (and to  $k_{ipso}$ , when appropriate), such that:

(5)"

**$k_{add} = \Sigma k F(\Phi) R(\Phi)$**

**Comment A8**: Page 8, line 5: So you totally ignore the aromatic carbons and use the estimated rate for the alkenyl group?

Response: We are very grateful to the referee for alerting us to this omission. As stated earlier in the section, addition to the aromatic ring is assumed to be completely deactivated in styrenes, based on a number of reported studies. However, for alkenyl-substituted aromatics containing more remote

C=C bonds, this would not be expected, although there are apparently no data to test this. The relevant text has therefore been adjusted as follows:

"The addition of OH to more remote C=C bonds in substituent groups in alkenyl-substituted aromatic hydrocarbons is expected to be well described by the methods described in the companion paper (Jenkin et al., 2018a), which update and extend the methods reported by Peeters et al. (2007) for alkenes and dienes. However, there are currently no data to test this assumption. In these cases, it is suggested that a default value of  $R(\Phi) = 1.0$  for the remote alkenyl group is applied for addition of OH to the aromatic ring."

**Technical corrections/comments:**

Page 5, line 18: Can't see that you defined kcalc , assume it is the same as k, defined as k=kadd+kabs (page 3, line 17)

Response: The referee is correct. The definition on page 3 has been amended to " $k_{calc} = k_{add} + k_{abs}$ ."

Consider replacing the "." in equations with "." or "x" to signify multiplication. It looks like a period.

Page 8, line 2: replace reference to Table 3 with Table 6.

Page 9, line 8: replace "upper panel" with "large panel" or "main panel".

Response: We are very grateful to the referee for identifying the above typos and technical corrections, which have all been corrected in the revised manuscript.

**B.** Comments by Referee 2**

**General comments:**

This manuscript describes the development of a structure-activity relationship (SAR) model for the reactions of OH with aromatic organic compounds which is explicit in terms of the OH reaction mechanism, and in subsequent reactions with molecular oxygen. These features likely will assist in the development of more detailed and quantitively correct representations of the atmospheric oxidation mechanisms for aromatic compounds. The work is carefully planned and performed, and the topical nature of the work makes it quite appropriate for publication in the Atmospheric Chemistry and Physics. The supporting information is very through and includes all experimental and calculated rate constant values, as well as examples to help the reader calculate rate constants from the SAR parameters.

Response: We are very grateful to the referee for these very positive and supportive comments on our work.

Specific comments

**Comment B1**: p.2: It would be good to note as a motivation that there really isn't a generalizable OH + aromatic SAR currently available in the literature.

Response: We thank the referee for this suggestion. This point has now been made in the description of the scope of the paper in the introduction:

"..... In each case, the rate coefficient is defined in terms of a summation of partial rate coefficients for H atom abstraction or OH addition at each relevant site in the given organic compound, so that the attack distribution is also defined. This is therefore the first generalizable SAR for reactions of OH with aromatic compounds that aims to capture observed trends in rate coefficients and the site-specificity of attack. Application of the methods is illustrated with examples in the Supplement."

**Comment B2**: p. 5, line 11: I assume from this discussion that the regression didn't use the experimental uncertainties in the rate constants to weight the individual values. Was anything done to take into account that the experimental rate constants have varying uncertainties?

Response: The referee is correct that different uncertainties were not assigned to the contributory preferred rate coefficients used in the analysis. We are also not aware of this being done in previous SAR development studies (e.g. Kwok and Atkinson, 1995; Calvert et al., 2008; 2011; Peeters et al., 2007). In practice it is very difficult to assign objective compound-specific uncertainties, because

most preferred values are derived from a number of contributing studies (some absolute and some relative rate) for which the quoted rate coefficients do not themselves have uncertainties reported consistently, such that a subjective judgement is required. The relative rate determinations are of course also influenced by uncertainties in the value of the reference rate coefficient. Using the IUPAC preferred values for benzene and toluene at 298 K as examples, these are each judged to have an uncertainty of a factor of about 1.25 by the IUPAC Task Group, although this is not based on a rigorous statistical analysis1. The former preferred value is based on the unweighted average of 9 determinations (all absolute) and the latter on the unweighted average of 11 determinations (5 absolute and 6 relative rate). Based on inspection of the preferred values for the other methyl-substituted aromatics (and the contributing studies), we judge that these determinations are unlikely to have uncertainties significantly greater than a factor of 1.25, such that a standard unweighted least squares analysis is justifiable.

**Comment B3**: p. 5, line 13: The comment about ortho- and para-substituents being more activating than metasubstituents is only true for the specific case of electron donating substituents such as methyl groups, which is also a well-known property of electrophilic aromatic substitution reactions.

Response: We thank the referee for this clarification.

**Comment B4**: Various tables: Why aren't uncertainties given for the various F(phi) values determined from the fitting process?**

Response: In common with all previous SAR development studies for atmospheric reactions that we are aware of (e.g. Kwok and Atkinson, 1995; Peeters et al., 2007; Calvert et al., 2008; 2011; Ziemann and Atkinson, 2012), we have chosen not to report uncertainties in the optimized parameters. This is because the calculation of a rate coefficient generally requires the use of several parameters, the values of which are not independent. Thus, it is not valid to vary the applied value of a given parameter within its uncertainty bounds, without making a compensating change in another parameter – and an assessment of the overall uncertainty in the final value of  $k_{calc}$  using combination and propagation of errors would not give a reliable estimate. Thus, although the optimized parameters are subject to uncertainties, these are generally not of practical value in applying the SAR. In practice, the performance of a SAR is mainly governed by the assumptions in the model framework that forms its basis, and the optimized parameters simply specify how to get the best performance out of the method within the constraints of the model framework. This performance is therefore generally assessed and improved by testing and refining the model and optimised parameters as the kinetics database expands and improves.

**Comment B5**: p. 6 line 16: I don't understand the problem being described here. From the statement earlier in this paragraph, I thought the H-abstraction values were being determined from p-cymene (the only compound for which H-abstraction experimental information is available), so I don't understand why these parameters then need to be adjusted.

Comment B6: Additionally, on what theoretical grounds might these adjustments be justified?

Response: We do not fully understand the point the referee is making. With reference to the parameters in Table 1, the relevant paragraph is explaining that the neighbouring group parameter F(-Ph1) optimized for H atom abstraction from  $-CH_3$  substituents does not give a good description of the reported contribution of H atom abstraction from the *i*-propyl substituent in *p*-cymene (and likely other larger substituents); and that it is therefore necessary to define a further parameter, F(-Ph2), for H atom abstraction from  $-CH_2$ - and -CH< substituents, based on the *p*-cymene data. Although F(-Ph2) is being introduced as a new parameter on this basis, there is no subsequent adjustment being described.

**Comment B7**: p. 6 line 30: The equation for R(phi) should be explicitly given.

<sup>1 http://iupac.pole-ether.fr/htdocs/supp\_info/Guide\_to\_Gas-Phase\_Datasheets\_Final\_Oct\_2017.pdf

Response: We thank the referee for this suggestion (see also comment A7 of referee 1). We have now formalised the method at this point, as suggested:

"Table 6 shows a set of adjustment factors for non-methyl substituents,  $R(\Phi)$ , that represent corrections to the values of  $F(\Phi)$  in Table 3 (and to  $k_{ipso}$ , when appropriate), such that:  $k_{add} = \sum k F(\Phi) R(\Phi)$  (5)"

**Comment B8**: Table 5: I understand that previous reports used different definitions for the branching ratio, but it is quite distracting and confusing to have two sets of values reported. I suggest that the authors convert all branching ratios to a common definition and to report that single set of values.

Response: We thank the referee for this suggestion. In the revised manuscript, we present all branching ratios relative to  $k_{add}$  in Table 5. Where the original reference reports values relative to  $k_{add} + k_{abs}$ , we also give the reported values in a footnote.

Comment B9: The form of equation 5 should be justified in the text.

Response: We agree with the referee's comment, and provide more explanation of the form of this equation in the revised manuscript. Following submission of the manuscript, we also realised that the previously declared equation (now Eq. (7)) did not describe the applied method for the case of n = 0, and we apologise for this omission. The relevant material has therefore been changed to read as follows:

"The value of  $k_{abs-O2}$  is assumed to be independent of the presence of alkyl substituents, but the value of  $k_{add-O2}$  depends on both the degree and distribution of alkyl substituents, and is given by:

(6)

(7)

 $k_{\text{add-O2}} = k^{\circ}_{\text{add-O2}} \prod F_{i}(X)$ , for n = 0 (or 1)

 $k_{\text{add-O2}} = k^{\circ}_{\text{add-O2}} \prod F_{i}(X)/n^{0.5}$ , for  $n \ge 1$

Here, *n* is the number of alkyl substituents (in positions 1 to 5 relative to the addition of  $O_2$ ), and  $F_i(X)$  is the activating effect of each alkyl substituent in terms of its position (see Fig. 3). The assigned values of  $F_i(X)$  (given in Table 8) recreate the reported general trend in total hydroxyarene yields for methyl-substituted aromatics, and also a reasonable representation of the reported distribution of isomers formed from a given aromatic precursor (see Table S1). In the case of the toluene system, for example, the optimized parameters provide respective yields of 12.2 %, 3.7 % and 3.3 % for *o*-, *m*- and *p*-cresol, and a total rate coefficient of 5.7 × 10-16 cm3 molecule-1 s-1 for the reaction of  $O_2$  with the set of OH-toluene adducts (i.e.  $HOC_7H_8$ ) at 298 K; in very good agreement with the IUPAC recommendations (IUPAC, 2017c). To a first approximation, the simpler expression in Eq. (6) provides an acceptable description for the complete series of aromatics, but leads to a systematic underestimation of the hydroxyarene yields reported for *m*-xylene, *p*-xylene, 1,2,4-trimethylbenzene and 1,3,5-trimethylbenzene. The adjusted expression in Eq. (7) is therefore defined to allow a more precise description of the reported hydroxyarene yields for the more substituted species."

Eq. (6) shows that  $k_{add-O2}$  is simply determined from the product of the reference value,  $k^{\circ}_{add-O2}$  (defined earlier), and the value of  $F_i(X)$  for each alkyl substituent. The need for the extra term,  $n^{0.5}$ , in Eq.(7) is now explained and justified in the new text.

**Comment B10**: Table S3: The authors should use the term "calculated" rather than "estimated" to be consistent with the other instances where rate constants calculated from the SAR model are reported. I understand that the red font entries in the "recommended" column are experimental values, but for the non-red font entries, what is the process for the determination of these recommended values?

Response: We thank the referee for pointing out this inconsistency, which has been corrected in the revised manuscript. Regarding explanation of the recommended values, the final column in Table S3 refers the reader to footnotes at the end of the table which explain how both the red font and non-red font entries were assigned. The non-red font channel contributions mainly retain the calculated relative importance, but with their absolute contributions reduced to account for the (minor) residual not covered by the yields of the products reported in experimental studies.

**Comment B11**: Figure 7: To what extent is the lower uncertainty evident in the aromatic set of compounds as compared to the aliphatic compounds a function of larger structural/functional group heterogeneity of the aliphatic compound group? Or is there another explanation?

Response: As stated by the referee, the lower uncertainty for the aromatic species reflects that the larger database of aliphatic compounds contains a more diverse set of oxygenated species, such that additional data for aromatic oxygenated species would be valuable. We believe this is already covered by the text in Sect. 6, where Fig. 7 is referred to:

".....This shows a similar pattern to that reported previously for the much larger dataset of aliphatic species (Jenkin et al., 2018a), but with systematically lower errors. As described in Sect. 3.2, some of the classes of aromatic oxygenated species contain data for only a single compound, such that the optimized parameters inevitably provide a good description of the observed data; whereas the aliphatic data are typically comprised of larger and more diverse sets of species. Additional rate coefficients would therefore be highly valuable for further assessment and evaluation of the SAR for a variety of aromatic oxygenated species."

**Technical corrections:**

Equation 4: The product sign between the two terms in the summation argument looks more like a decimal point. I suggest removing it entirely.

Response: We agree with the referee's suggestion and have removed points in equations throughout the revised manuscript.

**B.** Comments by Luc Vereecken (commenter)**

**Opening comment:**

This is an excellent paper, summarizing many aspects of aromatic chemistry. I will not comment on most of it, as I generally agree with how the analysis is done. Aromatic chemistry is very complex, and the authors are the first to propose useable SARs that capture rate coefficient and site-specificity trends.

Response: We are very grateful to Luc Vereecken for these very positive and supportive comments on our work, and for submitting a detailed and informative discussion comment.

There is, however, one aspect I feel is not in agreement with the experimental observations and the theoretical data available. In particular, I have reservations on a mechanism that incorporates the chemically activated "peroxide-bicyclic" radical (**BCP-yl** in figure LV-1) as an important source of up to 30% of the products. My reasoning is described below in several parts; it is obviously based on incomplete data and thus by no means final. Many of my remarks are based on my recent overview of theoretical studies on aromatic chemistry, of which the authors have received a copy (though much too late to incorporate that data in the paper discussed here). I have summarized the chemical mechanism in figure LV-1 below, and will use the naming and reaction labels in that figure.

I recognize that the proposed mechanism (Figure 3 and 4 in Jenkin et al. 2018) is only a small part of the paper, and this comment is thus not a major criticism of that work. My worry is mainly that no systematic mechanistic improvement can be done by updating rate coefficients and yields of elementary reactions, if the mechanism in the model does not match the underlying chemical process. Given that this mechanism would be implemented in the MCM, the most commonly used semi-explicit mechanism, this could hamper progress significantly. This comment is not as complete as I wanted it to be and may contain errors, as I ran out of time trying to meet the deadline for comment submission. I apologize for the poor presentation of this text and the lack of a more thorough numerical analysis, and remain available to clarify this text.

Figure LV-1 : Extended mechanism for aromatic oxidation (toluene used as example). Stereo- and site-specificity is ignored.

Response: We agree that the commenter has raised an area of particular uncertainty in understanding, and welcome this detailed feedback and discussion to allow this to be further highlighted and emphasised. Following careful consideration of the subsequent comments, and the information in the cited literature, we have decided to leave our method unchanged at the present time (as discussed and justified further below). However, we have given additional emphasis to this area of uncertainty in the revised manuscript, including reference to the commenter's discussion comment and to his highly informative forthcoming review of theoretical studies on aromatic oxidation. We would like to emphasise that we have not dismissed the commenter's concerns lightly, and recognise the validity of those concerns, and the insight that has gone into the mechanistic interpretation he has put forward. We hope that advances in understanding will soon allow the various issues discussed below to be reconciled.

We also recognise the commenter's point about processes becoming "hard-wired" into the MCM, and agree that updating the mechanism efficiently has been challenging in the past. A main aim of the current move to automated mechanism construction is to allow updates in understanding to be implemented into the mechanism more readily and efficiently.

**a) Chemical activation for BCP-yl.**

While it is clear that the nascent **BCP-yl** must have a high internal energy, there is no evidence that it reacts chemically activated to any significant extent, and to me seems unlikely that 30% of the **BCP-yl** will isomerise promptly to epoxy-oxy radicals (**BCE-O**) even for the smallest aromatics.

Prompt decomposition of intermediates in aromatic oxidation, or the pressure dependence of the aromatic oxidation has been studied by several authors ((Glowacki et al., 2009; Lay et al., 1996; Mehta et al., 2009; Pan and Wang, 2015; Wu et al., 2014). Under atmospheric conditions, i.e. around 1 atm. of pressure and air as a bath gas, these studies consistently find that prompt decomposition (channel **E**) has no significant yield. For benzene, Glowacki et al. find at most a few % of prompt decomposition of **BCP-yl** to **BCE-O**, and increasing this yield to 30% would require strong modifications of the kinetic model, likely beyond the reasonable error limits of the applied theoretical methodology. Substituted aromatics have an even higher number of degrees of freedom to redistribute essentially the same nascent excess energy, such that high prompt isomerisation yields of **BCE-O** become even less likely for the more substituted aromatics in the atmosphere. There is no evidence of lower epoxide formation for more substituted reactions.

In this, one should account for the fact that the theoretical studies such as Glowacki et al. did not consider the bath gas as a reactive collider, i.e. their already low yields of prompt **BCE-O** formation do not account for the near-barrierless addition of  $O_2$  (1/5th of the bath gas collisions), onto **BCP-yl**, forming **BCP-OO** peroxy radicals prior to thermalization, and thus further reducing the yield of BCE-O. One could argue that the energized **BCP-OO** would also be more likely to redissociate, but there will always be sufficient time for energy randomization, such that the leaving  $O_2$  fragment (and the degrees of freedom for relative motion of the fragments) would remove above-thermal energies from **BCP-yl**, leading to even more efficient collisional cooling of the activated **BCP-yl**, and hence smaller contributions of ring opening than the already small predicted yields.

All theoretical calculations indicate that the barrier for peroxide-ring breaking in **BCP-yl** has high barriers across all substituent patterns of aromatics, and the low contribution of prompt ring opening thus appears to hold for all aromatics ((Fan and Zhang, 2006, 2008; Glowacki et al., 2009; Huang et al., 2008, 2010; Li and Wang, 2014; Pan and Wang, 2014, 2015; Suh et al., 2003; Wang, 2015; Wu et al., 2014; Xu and Wang, 2013).

Response: We agree that the 30% proportion we assign to decomposition of BCP-yl is higher than that reported for atmospheric pressure in theoretical studies, and this point was made in the manuscript – although not as clearly as it could have been. We have therefore further emphasised this in the revised manuscript, with reference to the commenter's discussion comment and to his forthcoming review. The relevant text in the manuscript has been modified to read as follows (revised/new text in red font):

"Inclusion of the "epoxy-oxy" route with this optimized branching ratio results in total prompt HO2 yields which provide a good representation of those reported by Nehr et al. (2011; 2012), and also the total yields of the well-established  $\alpha$ -dicarbonyl products (formed from the alternative O2 addition chemistry) that are consistent with those reported (see below). However, it is noted that this is an area of significant uncertainty, with theoretical studies predicting a much lower importance of the "epoxy-oxy" route at atmospheric pressure than applied here (e.g. Vereecken, 2018a; 2018b; and references therein). Further studies are required to elucidate the sources of epoxydicarbonylenes and prompt HO2 in aromatic systems."

As indicated in the manuscript, and discussed further below, the figure of 30 % was empirically optimised on the basis of reported yields of a variety of products (particularly  $\alpha$ -dicarbonyls and prompt HO2) for a series of aromatic hydrocarbons, with evidence for formation of the epoxydicarbonylene products (e.g. epoxy-MHDD in Fig. LV1) formed from BCP-yl decomposition being reported in a number of experimental studies (as cited). In our opinion, a significant reduction in this proportion would result in the mechanism failing to reproduce quantitatively the majority of reported experimental observations. While we accept that this does not prove that the assignment of 30 % to this specific process is correct, the use of a reaction for which there is at least some experimental (and indeed theoretical) support is considered an acceptable interim measure until alternative quantitative explanations are available that do not degrade other aspects of the mechanism's performance.

To illustrate this, Fig. R1 below shows a correlation of calculated and observed yields of the relevant species (a) with the optimised branching ratio of 30 % assigned to BCP-yl decomposition (i.e. as in Fig. 5 of the manuscript); and (b) with a branching ratio of 0 % assigned to BCP-yl decomposition: